# Comparison of kidney allograft survival in the Eurotransplant senior program after changing the allocation criteria in 2010—A single center experience

Anne-Sophie Mehdorn[1], Stefan Reuter[2]*, Barbara Suwelack[2], Katharina Schütte-Nütgen[2], Felix Becker[1], Norbert Senninger[1], Daniel Palmes[1], Thomas Vogel[1], Ralf Bahde[1]

1 Department of General, Visceral and Transplant Surgery, Münster University Hospital, Münster, Germany,
2 Department of Medicine D, Division of General Internal Medicine, Nephrology and Rheumatology, Münster University Hospital, Münster, Germany

* sreuter@uni-muenster.de

**Data Availability Statement:** All relevant data are within the paper.

## Abstract

### Aims

The European Senior Program (ESP) aims to avoid waiting list competition between younger and elderly patients applying for renal transplantation. By listing patients ≥65 years on a separate waiting list and locally allocating of grafts ≥65 years exclusively to this cohort, waiting and cold ischemia times are predicted to be shortened, potentially resulting in improved kidney transplantation outcomes. This study compared a historic cohort of renal transplant recipients being simultaneously listed on the general and the ESP waiting lists with a collective exclusively listed on the ESP list in terms of surrogates of the transplantation outcome.

### Methods

Total 151 eligible patients ≥ 65 years from Münster transplant Center, Germany, between 1999 and 2014 were included. Graft function, graft and patient survival were compared using surrogate markers of short- and long-term graft function. Patients were grouped according to their time of transplantation.

### Results

Recipients and donors in the newESP (nESP) cohort were significantly older (69.6 ± 3.5 years vs 67.1 ± 2 years, p<0.05; 72.0 ± 5.0 years vs 70.3 ± 5.0 years, p = 0.039), had significantly shorter dialysis vintage (19.6 ± 21.7 months vs 60.2 ± 28.1 months, p<0.001) and suffered from significantly more comorbidities (2.2 ± 0.9 vs 1.8 ± 0.8, p = 0.009) than the historic cohort (HC). Five-year death-censored graft survival was better than in the HC, but 5-year graft and patient survival were better in the ESP cohort. After 2005, cold ischemia time between groups was comparable. nESP grafts showed more primary function and significantly better long-term graft function 18 months after transplantation and onwards.

**Funding:** This work was funded by the Open Access Publication Funds of the University of Münster.

**Competing interests:** The authors have declared that no competing interests exist.

## Conclusion

nESP recipients received significantly older grafts, but experienced significantly shorter time on dialysis. Cold ischemia times were comparable, but graft function in the nESP cohort was significantly better in the long term.

## Background

Kidney transplantation (KTx) is the only curative treatment for end-stage renal disease (ESRD), improving quality of life and life expectancy compared to renal replacement therapy (RRT) [1]. Currently, patients >65 years represent the fastest growing population on the transplantation waiting list [2–4]. As no cut off-age exists, KTx has even been performed in 80-year-old patients [1, 5]. However, recently, the discrepancy between organ demand and supply has further escalated: a decreasing number of organ donations meets an increasing number of patients waiting for KTx, therefore an expansion of the donor pool is necessary [6–9]. Extended donor criteria (ECD) include brain death donors ≥60 years or donors aged 50–59 years who suffered from at least two comorbidities: hypertension, elevated serum creatinine (>1.5 mg/dl) or death from cardiovascular diseases [10]. However, these grafts are considered marginal due to their age, previous lifetime and associated comorbidities [11]. The potentially poorer quality and impaired recovery capacity can result in reduced graft function and survival [1, 12].

To meet this demand, Eurotransplant (ET) introduced the European Senior Program (ESP) in addition to the ET Kidney Allocation System (ETKAS) procedure in 1999 [13]. It moved the focus from an immunological Human Leukocyte Antigen (HLA)-based allocation to an allocation procedure that emphasizes age and cold ischemia time (CIT) [14]. Local allocation of grafts ≥65 years to non-immunized elderly recipients (≥65 years) without taking HLA-matching into account should keep CIT as short as possible [13]. Further, competition between elderly and younger patients for grafts and waiting times should be minimized [3, 13]. During the first years, eligible patients ≥65 years were listed for both programs. Since 2010, though, patients have had to pick a program as both programs became mutually exclusive [13]. Decision for one program was made after discussion with the consulting transplant nephrologist. It was hypothesized that the changes introduced by the ESP would result in better graft function and survival, thereby outweighing the age-associated disadvantages of the elderly graft [12, 15]. First analysis of the ESP showed promising results [16, 17]. Further, Frei et al. could not identify a negative influence of elderly grafts in similarly aged or younger recipients in an initial 5-year analysis [18]. However, their data did not include the years after 2010 when the choice for one waiting list became mandatory [18]. Therefore, this study aims to determine whether a single listing and the exclusive allocation of elderly ESP-grafts puts elderly patients at a disadvantage or even penalizes them compared to a cohort listed on both waiting lists.

## Material and methods

### Study design

Similar to a retrospective study previously published from Münster Transplant Center (MTC), two cohorts after KTx were compared [19]. The first cohort included patients receiving KTx according to the new ESP-policy between October 2010 and September 2014 (newESP cohort,

nESPC). The second was an historic cohort (HC) which underwent KTx between January 1999 and May 2007 after parallel listing on the ETKAS and the ESP waiting lists [19]. However, only patients who received an ESP kidney were included in the final analysis. Recipient data included demographic data, underlying renal disease, most common comorbidities and time of dialysis vintage [19–21]. The dialysis vintage served as surrogate for the recruiting period. To further characterize RTx-recipients, comorbidities, including arterial hypertension, diabetes mellitus, hyperlipidemia and arteriosclerosis as well as BMI as a surrogate for overweight were included in the analysis. Arterial hypertension and diabetes mellitus are well described major risk factors in patients with chronic kidney diseases as are hyperlipidemia and overweight [20]. Arteriosclerosis may influence surgical outcome and was therefore included as well [21]. Ethics committee of the University of Münster and local ethics committee have approved and provided written ethical approval (No. 2014-381-f-N). The Declaration of Helsinki served as the ethical base for the study. Prior to surgery, all patients gave their written consent for the recording and use of their clinical data. ESP-enlisted patients had separately given informed written consent for their enrolment in the ESP. Recipient and donor data were extracted from Eurotransplant (ET) files and from the clinical database and de-identified prior to analysis. Patients' eligibility, surgical procedure and postoperative immunosuppressive treatment were as described previously [19]. Briefly, standard KTx was performed with an extra-peritoneal preparation, end-to-side-vascular anastomosis to the iliac vessels if possible and a modified anti-reflux ureterocystostomy according to Lich-Gregoire. Until 2007, the ureterocystostomy was protected by an externalized uretro-vesico-cutaneous stent and a suprapubic catheter for ten days. In the nESPC, the ureteral anastomosis was splinted by a double J-catheter scheduled to stay for 6 weeks. Patients received a transurethral catheter for five days. Pre-transplant AB0-blood group compatibility, negative PRA-antibodies and negative cross match were mandatory, while HLA-matching was abandoned [13]. According to MTC's protocol, standard immunosuppression consisted of tacrolimus, mycophenolate mofetil and steroids. Basiliximab-induction at days 0 and 4 started in 2006. Until 2003, cyclosporine A was part of the immunosuppressive therapy, but was later abandoned. In cases of cytomegalovirus-IgG positivity, CMV prophylaxis with valganciclovir was administered. Co-trimoxazole was administered for Pneumcystis carinii-prophylaxis. All donors were heart-beating, brain dead donors aged ≥65 years. No donations from prisoners, minor donors or people with reduced mental health were included. Informed consent for organ donation was obtained according to German organ donation law. Patients decided during their lifetime if they wanted to become organ donors. If patients were considered eligible for donation and had not given written consent prior to their death, close relatives decided on the patient's behalf after the patient's death acknowledging the potential will of the deceased.

## Endpoints

The primary endpoint was graft function using serum creatinine and estimated glomerular filtration rate (eGFR) calculated according Modification of Diet in Renal Disease Study Group (MDRD) as surrogates [22]. Graft function was chosen as primary endpoint due to the small number of patients included. Primary function (PF), delayed graft function (DGF) and primary non-function (PNF) were defined according to the surrogates and postoperative need for dialysis. DGF was assumed in cases of dialysis within the first week post transplantation. PNF was assumed if the graft did not show signs of recovery after re-initiation of dialysis.

Secondary endpoints were graft and patient survival. Patient survival was defined as time from KTx to death from any cause or last contact with patient alive. Graft survival was defined accordingly from KTx to death from any cause, graft failure or last contact whichever occurred

first. Graft failure was defined as re-initiation of dialysis treatment. Death-censored graft survival was defined as time from KTx to graft failure. Death without previous graft failure was regarded as censored. Further, demographic data including underlying renal disease, comorbidities, time of dialysis, HLA mismatch, cold and warm ischemia times, intra- and postoperative complications and biopsy-proven rejection (BPR) were analyzed and compared and their influences on graft outcome tested.

## Statistics

Quantitative data was presented as percentages and evaluated using the Student's t-test. Qualitative data was expressed as means ± standard deviation and ranges and was evaluated using Fisher's exact test. P-values <0.05 were considered significant. Cox regression was used to evaluate the influence on 5-year death-censored graft function and survival and 5-year patient survival [23]. Hazard's ratio, corresponding 95%-confidence intervals and p-values were reported. For multivariable model building, variables with a p-value of the likelihood ratio test > 0.1 were excluded and only variables with a p-value less than 0.1 were further analyzed by multivariable regression. Survival data of graft and patient were interpreted by Kaplan-Meier method and further evaluated by log-rank test [24]. SPSS version 24 was used for statistical analysis (SPSS Inc., Chicago, IL, USA).

## Results

### Study population

Between October 2010 and September 2014, 62 elderly patients received a KTx according to the new ESP-allocation scheme (nESPC). Between January 1999 and May 2007, 89 elderly patients had transplants after being listed on both waiting lists (HC) [19]. Yet, only recipients of an ESP-kidney were included in the final analysis. Patients in the nESPC were older (69.6 ± 3.5 years vs 67.1 ± 2.6 years, p < 0.001), heavier (28.5 ± 5.5 kg/m$^2$ vs 25.3 ± 3.4 kg/m$^2$, p < 0.001), more often male (74.2% vs 69.7%, ns) and sicker (2.2 ± 0.9 comorbidities vs 1.8 ± 0.8 comorbidities, p = 0.009). Dialysis vintage was significantly shorter in the nESPC (19.6 ± 21.7 months vs 60.2 ± 28.1 months, p = 0.009). In the nESPC 15 patients died while waiting for an ESP-kidney. For the HC exact numbers cannot be reconstructed. In a comparable cohort from Münster Transplant Center, 5-year survival on the waiting list was only 41% [25]. Chronic glomerulonephritis was the primary reason for ESRD in both groups (32.3% vs 43.8%) (Table 1). Seven nESPC recipients and 12 HC-patients had previously undergone KTx before receiving an ESP kidney. A total of 68 HC-patients received an induction therapy with a monoclonal IL-2 receptor antibody (basiliximab) and the most common (used for 47 patients) immunosuppressive maintenance regime was triple therapy consisting of cyclosporine, mycophenolate mofetile and steroids until 2003. Hereafter, maintenance regimes consisted of tacrolimus, mycophenolate mofetil and steroids. The average follow-up period was 1,039.2 ± 497.8 days (nESPC) and 839.4 ± 704.6 days (HC). The 5-year overall follow-up was 20.8% and 14.8% (nESPC) and 26.4% (HC), respectively.

Fifty nESPC donors were compared to 62 HC donors. Local allocation allowed MTC to receive both kidneys from 14 donors in the nESPC and 12 in the HC. However, patients only received single kidney transplantation. nESP-recipients after 2010 were not part of the Eurotransplant Senior DR-compatible Program (ESDP) initiative of Eurotransplant. nESPC donors were older (72.0 ± 5.0 years vs 70.3 ± 5.0 years, p = 0.039) and more often male (59.7% vs 38.2%, p = 0.013). Weight (26.4 ± 3.9 kg/m$^2$ vs 27.5 ± 3.6 kg/m$^2$, p = 0.055), kidney function pre-procurement (serum creatinine 1.0 ± 0.4 mg/dl vs 1.0 ± 0.5 mg/dl, ns); diuresis/last 24 hours (4,074 ± 2,387 ml vs 4,058 ± 2,226 ml, ns) and the number of comorbidities were

**Table 1. Baseline comparison of kidney transplant recipient stratified by cohorts (nESPC vs HC).**

| Recipient characteristics | | | |
|---|---|---|---|
| | New ESP cohort (n = 62) | Historic cohort (n = 89) | p-value |
| **Age** (years, mean ± SD, range) | 69.6 ± 3.5 (65–78) | 67.1 ± 2.6 (65–79) | **<0.001[a]** |
| **Sex** (% male) | 74.2 | 69.7 | 0.586[b] |
| **BMI** (kg/m², median ± SD, range) | 28.5 ± 5.5 (18–49) | 25.3 ± 3.4 (17–35) | **<0.001[a]** |
| **BMI > 25kg/m²**(%) | 74.2 | 55.1 | **0.018[b]** |
| **Indication for transplantation** (%) | | | **0.011[c]** |
| Chronic glomerulonephritis | 32.3 | 43.8 | |
| Nephrosclerosis | 19.4 | 10.1 | |
| ADPKD | 16.1 | 14.6 | |
| Diabetic nephropathia | 16.1 | 4.5 | |
| Interstitial nephritis | 1.6 | 9.0 | |
| Renal malignoma | 3.2 | 3.4 | |
| Chronic pyelonephritis | 0 | 6.7 | |
| Other | 9.7 | 3.4 | |
| Unknown | 1.6 | 4.5 | |
| **Time of RRT** (months, mean ± SD, range) | 19.6 ± 21.7 (1–90) | 60.2 ± 28.1 (11–146) | **<0.001[a]** |
| **Sum of comorbidities** (mean ± SD) | 2.2 ± 0.9 | 1.8 ± 0.8 | **0.009[a]** |
| **Comorbidities** (%) | | | |
| Hypertension | 93.5 | 73.0 | **0.001[b]** |
| Diabetes | 35.5 | 13.5 | **0.003[b]** |
| Arteriosclerosis | 56.5 | 73.0 | **0.038[b]** |
| Hyperlipidemia | 24.2 | 16.9 | 0.303[b] |

Data are presented as mean ± standard deviation (SD), median, min and max or relative frequencies. Categorical variables were compared using Fisher's exact test while continuous variables were compared using Student's t's t-test (normally distributed) or Mann-Whitney U test (not normally distributed). ESP = European Senior Program, BMI = body mass index, ADPKD = Autosomal-dominant polycystic kidney disease, RRT = renal replacement therapy a) Student's t-test, b) Fisher's exact test, c) Pearson's square test. p-values less than 0.05 were considered statistically significant.

comparable. Nevertheless, nESPC donors suffered more frequently from hypertension (48.4% vs 28.1%, p = 0.016), while HC donors had more frequently arteriosclerosis (16.1% vs 48.3%, p<0.001) (Table 2). Intracranial bleeding was the primary cause of brain death in both cohorts. Age difference between donors and recipients was reduced from 2.5 ± 0.9 years in the HC to 1.7 ± 0.1 years in the nESPC (ns).

A significant overall reduction of CIT was achieved by the introduction of the nESP procedure (9:14 ± 3:4 hours vs 11:7 ± 5:0 hours, p<0.001), while HLA mismatches remained comparable (4.3 ± 1.2 vs 4.1 ± 1.1, ns). However, changes in the allocation procedure after 2005 reduced CIT in the HC from 14:85 ± 4:3 hours to 8:27 ± 3:04 hours (p = 0.022). Thereafter, both cohorts experienced comparable CIT (8:27 ± 3:04 hours vs 9:14 ± 3:54 hours, ns) (Table 3).

## Primary endpoints

Overall graft function and 1-, 3- and 5-year graft survival was comparable in both cohorts. However, the nESPC tended to be more likely to achieve primary function. The number of non-functioning grafts was slightly higher in the HC (Table 4). Immediately post transplantation, the function of HC grafts was better than that of grafts of the nESPC (creatinine: 5.4 ± 2.1 mg/dl vs 3.1 ± 2.4 mg/dl, p<0.001, eGFR: 11.7 ± 7.2 ml/min vs 22.6 ± 17.1 ml/min, p<0.001).

**Table 2. Baseline comparison of kidney donors stratified by cohorts (nESPC vs HC).**

| Donor characteristics | | | |
|---|---|---|---|
| | New ESP cohort (n = 50) | Historic cohort (n = 62) | p-value |
| **Age** (years, mean ± SD, range) | 72.0 ± 5.0 (65–84) | 70.3 ± 5.0 (63–68) | **0.039[a]** |
| **Sex** (% male) | 59.7 | 38.2 | **0.013[b]** |
| **BMI** (kg/m², mean ± SD, range) | 26.4 ± 3.9 (13–40) | 27.5 ± 3.6 (17–35) | 0.055[a] |
| **Days on ICU** (days, mean ± SD) | 3.9 ± 4.0 (1–25) | 4.8 ± 4.3 (1–18) | 0.218[a] |
| **Cause of death** (%) | | | 0.075[c] |
| Intracranial bleeding | 43.5 | 47.2 | |
| SAB | 19.4 | 29.2 | |
| Cerebral infarction | 22.6 | 10.1 | |
| Head trauma | 9.7 | 13.5 | |
| Hypoxic damage | 3.2 | 0 | |
| Cerebral edema | 1.6 | 0 | |
| **Creatinine pre-procurement** (mg/dl, mean ± SD) | 1.0 ± 0.4 | 1.0 ± 0.5 | 0.403[a] |
| **Diuresis/24h (ml)** | 4074 ± 2387 | 4058 ± 2226 | 0.860[a] |
| **Sum of comorbidities** | 0.9 | 0.9 | 0.976[a] |
| **Comorbidities (%)** | | | |
| Hypertension | 48.4 | 28.1 | **0.016[b]** |
| Diabetes | 21.0 | 9.0 | 0.054[b] |
| Arteriosclerosis | 16.1 | 48.3 | **<0.001[b]** |
| Hyperlipidemia | 4.8 | 4.5 | 1.00[b] |
| Age difference recipient–donor (y, mean ± SD)) | | 2.5 ± 0.9 | 0.689[c] |
| Age difference recipient–donor (y, mean ± SD) | 1.7 ± 0.1 | | 0.471[c] |

Data are presented as mean ± standard deviation (SD), median, min and max or relative frequencies. Categorical variables were compared using Fisher's exact test while continuous variables were compared using Student's t-test (normally distributed) or Mann-Whitney U test (not normally distributed). ESP = European Senior Program, BMI = body mass index, ICU = intensive care unit, SAB = subarachnoidal bleeding. a) Student's t-test, b) Fisher's exact test, c) Pearson's square test. p-values <0.05 were considered statistically significant.

However, three months after transplantation, creatinine levels were similar (1.8 ± 0.7 mg/dl vs 2.2 ± 1.5 mg/dl, ns) and 18 months post transplantation, creatinine levels in the nESPC were significantly lower (18 months: 1.7 ± 0.9 mg/dl vs 1.9 ± 0.5 mg/dl, p = 0.023). Creatinine remained significantly lower until 60 months after transplantation (60 months: 1.5 ± 0.3 mg/dl vs 1.9 ± 0.4 mg/dl, p = 0.024) (Fig 1).

The overall survival of patients was comparable between the cohorts. In contrast, death censored graft survival after three and five years was significantly better in the nESPC (Fig 2, Table 4). As the number of complete 5-year follow-ups was patchy, especially in the nESPC as patients have been transplanted more recently, the 1- and 3-year survival rates were calculated (Table 4). 3-year overall patient and graft survival did not differ significantly. Yet, 3-year death-censored graft survival was significantly better in the new cohort (p = 0.037). The primary causes for death with a functioning graft were sepsis, pneumonia, multi-organ failure and cardiac events. Furthermore, the main reasons for loss of function were biopsy-proven rejection (21.5% vs 19.1%, ns). Acute rejection episodes were in tendency increased in the nESPC (15.4% vs 13.5%, ns). Interestingly, only recipients' overweight was found to significantly influence 5-year graft survival in the multivariate analysis (p = 0.012) (Table 5). 5-year-death-censored graft survival was not influenced significantly by any factor in the multivariate analysis (Table 6).

**Table 3.** Summary of times and complications stratified by cohorts (nESPC vs HC).

| Times and Complications | | | |
|---|---|---|---|
| | New ESP cohort (n = 62) | Historic cohort (n = 89) | p-value |
| **Cold ischemia time** | 9:14 ± 3:35 | 11:74–4:99 | **<0.001[a]** |
| (hh:min, mean ± SD (min—max)) | (5:00–20:00) | (3:86–23:30)* | |
| **Cold ischemia time** | 9:14 ± 3.35 | 8:27 ± 3.03 | 0.241[a] |
| (hh:min, mean ± SD (min—max)) (2005–2014) | (5:00–20:00) n = 62 | (3:86–14:86) n = 42 | |
| **HLA mismatch** (n) | 4.3 ± 1.2 | 4.1 ± 1.1 | 0.325[a] |
| 'A' locus | 1.3 ± 0.6 | 1.2 ± 0.7 | 0.331[a] |
| 'B' locus | 1.6 ± 0.6 | 1.7 ± 0.5 | 0.427[a] |
| 'DR' locus | 1.4 ± 0.7 | 1.5 ± 0.5 | 0.523[a] |
| **Operation time** | 3:03 ± 0:57 | 2:47 ± 1:00 | 0.158[a] |
| (hh:min, mean ± SD, range) | (1:30–5:59) | (1:30–6:50) | |
| **Warm ischemia time** | 36.4 ± 9.8 | 41.7 ± 13.9 | **0.010[a]** |
| (min, mean ± SD, (min—max)) | (20–75) | (20–75) | |
| **Length of hospital stay** | 23.9 ± 16.2 | 25.2 ± 22.0 | 0.727[a] |
| (days, mean ± SD (min, max)) | (4–102) | (7–111) | |
| **Intraoperative complications** (%) | 3.1 | 15.7 | **0.283[c]** |
| Bleeding | 1.5 | 6.7 | 0.649[b] |
| Arteriosclerosis recipient | 3.1 | 6.7 | 0.701[b] |
| Graft injury during procurement | 0 | 2.2 | 0.513[b] |
| Cardiac arrest | 0 | 2.2 | 0.513[b] |
| **Postoperative complications** (%, yes) | 69.2 | 53.9 | 0.067[b] |
| **Postoperative surgical complications** (%) | 50.8 | 20.2 | **<0.001[c]** |
| Hematoma/Bleeding | 21.5 | 12.4 | 0.184[b] |
| Wound infection | 3.1 | 1.1 | 0.572[b] |
| Other infection | 1.5 | 2.2 | 1.000[b] |
| Ureteral anastomotic leakage/stenosis | 0 | 4.5 | 0.138[b] |
| Dehiscence of the fascia | 9.2 | 0 | **0.005[b]** |
| Lymphocele | 12.3 | 0 | **0.001[b]** |
| Hydronephrosis | 3.1 | 0 | **0.005[b]** |
| **Postoperative non-surgical complications** (%) | 49.5 | 33.7 | **<0.001[c]** |
| Cardiac complications | 10.8 | 7.9 | **0.022[b]** |
| UTI | 27.7 | 12.4 | 0.579[b] |
| Thrombosis of Graft's Vein | 0 | 13.5 | **0.001[b]** |
| Bone marrow toxicity | 10.8 | 0 | **0.002[b]** |

Data are presented as mean ± standard deviation (SD), median, min and max or relative frequencies. Categorical variables were compared using Fisher's exact test while continuous variables were compared using Student's t-test (normally distributed) or Mann-Whitney U test (not normally distributed). ESP = European Senior Program, HLA = human leukocyte antigen, UTI = urinary tract infections, SIDM = steroid-induced diabetes mellitus a) Student's t-test, b) Fisher's exact test, c) Pearson's square test. P-values <0.05 were considered statistically significant.

## Intraoperative and postoperative complications

Operation time in the nESPC was slightly longer (3:03 ± 0:57 hours vs 2:47 ± 1:00 hours, ns), but anastomosis time was significantly shorter (36:4 ± 9:8 min vs 41:7 ± 13:9 min, ns) (Table 3). Intraoperative complications occurred in 3.1% of the nESPC and in 15.7% of the HC (p = 0.014), respectively. Intraoperative bleeding, either due to graft injury during procurement or vascular problems, was the most common complication in both cohorts (Table 3). Surgical and non-surgical postoperative complications dominated in the nESPC (50.8% vs

**Table 4. Primary and secondary outcomes after kidney transplantation stratified by cohorts (nESPC vs HC).**

| Primary and secondary outcomes | | | |
|---|---|---|---|
| | New ESP cohort (n = 65) | Historic cohort (n = 89) | p-value |
| **Graft function** (%) | | | 0.240[c] |
| **PF** | 72.6 | 67.4 | 0.591[b] |
| **DGF** | 25.8 | 24.7 | 1.000[b] |
| **PNF** | 1.6 | 7.9 | 0.142[b] |
| **Overall graft survival** (%) | | | |
| 1 y | 87.1 | 79.3 | 0.215[a] |
| 3 y | 83.9 | 71.3 | 0.110[a] |
| 5 y | 82.3 | 62.1 | 0.053[a] |
| **Death-censored graft survival** (%) | | | |
| 1 y | 93.5 | 82.3 | 0.141[a] |
| 3 y | 93.5 | 33.9 | 0.037[a] |
| 5 y | 93.5 | 79.3 | 0.024[a] |
| **Loss of function** (%) | 26.2 | 23.6 | 0.113[c] |
| BPR | 16.9 | 10.1 | |
| PNF | 1.5 | 9.0 | |
| Renal vein thrombosis | 1.5 | 3.4 | |
| Recurrent urinary tract infection | 1.5 | 0 | |
| Medical intoxication | 1.5 | 0 | |
| Diabetic glomerulosclerosis | 0 | 1.1 | |
| Unknown | 3.1 | 0 | |
| **Patient survival** (%) | | | |
| 1 y | 93.5 | 92.0 | 0.672[a] |
| 3 y | 88.7 | 82.8 | 0.452[a] |
| 5 y | 87.1 | 73.6 | 0.223[a] |
| **Death with functioning graft** (%) | 12.3 | 16.9 | 0.147[c] |
| Sepsis/Pneumonia/MOF | 1.5 | 6.7 | |
| Cardiac events | 1.4 | 4.5 | |
| Refusal of immunosuppressive therapy | 1.5 | 0 | |
| Unknown | 9.2 | 5.6 | |
| Biopsy-proven rejection (%) | 21.5 | 19.1 | 0.932[c] |
| Acute | 15.4 | 13.5 | 0.817[b] |
| Chronic | 6.2 | 4.5 | 0.722[b] |

Results are presented as mean ± standard deviation (SD), median, min, max or relative frequencies. Categorical variables were compared using Fisher's exact test while continuous variables were compared using Student's t-test (normally distributed) or Mann-Whitney U test (not normally distributed). PF = primary function, DGF = delayed graft function, PNF = primary non-function, y = year, MOF = multi-organ failure. a) Student's t-test, b) Fisher's exact test, c) Mann-Whitney U test and d) Log-rank test, p-values <0.05 were considered statistically significant.

20.2% and 49.5% vs 33.7%, respectively, p<0.001). Bleeding was the main surgical complication in both cohorts (21.5% vs 12.4%, ns) followed by postoperative lymphocele in the nESPC (12.3%) and problems of urethral anastomosis in the HC (4.5%). Urinary tract infections (27.7% vs 12.4%, ns) and renal vein thrombosis (0% vs 13.5%, p<0.001) were the most common non-surgical complications. The occurrence of non-surgical complications influenced 5-year-patient-survival significantly in the multivariable analysis. (p = 0.029) (Table 7). The length of hospital stay was comparable (23.9 ± 16.2 days vs 25.2 ± 22.0 days, ns) (Table 3).

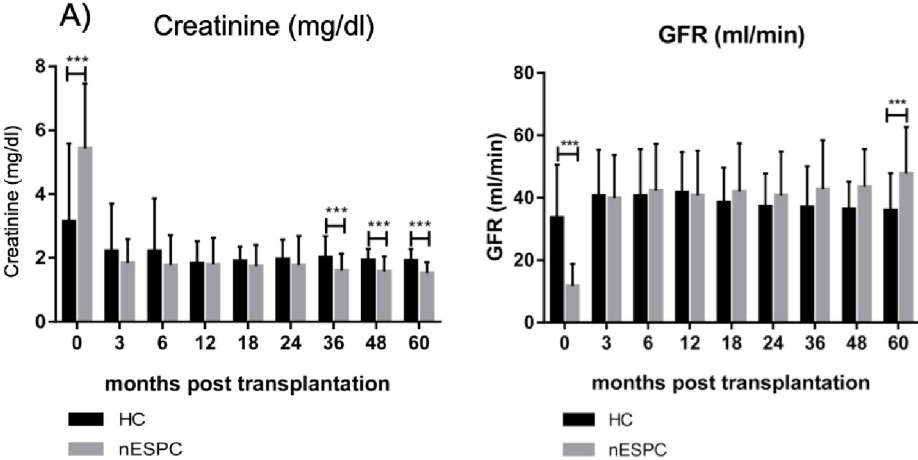

**Fig 1.** Kidney function using a) creatinine and b) GFR as surrogates during the follow-up period of 60 months. (black HC, grey nESPC). HC = historic cohort, nESPC = new European Senior Program cohort, *** highly significant, p < 0.001. Data are presented as mean ± standard deviation. Student's t-test was used to test for statistical significance. p-values <0.05 was considered statistically significant.

## Discussion

We hypothesized that the change from simultaneously ETKAS and ESP (HC) to mono-ESP (nESP cohort) listing would affect waiting time and outcome of KTx in elderly recipients. Patients in the nESP cohort were older and heavier, suffered more often from hypertension and diabetes, but had experienced shorter dialysis vintage and showed less arteriosclerosis. Yet, more surgical and non-surgical postoperative complications were reported in the nESP cohort. Despite older and sicker recipients as well as older donors in the nESPC, nESP grafts showed slightly more primary and a significantly better long-term function. Our study did not find a negative effect of the allocation of older grafts after the introduction of the new listing procedure at MTC. The new allocation procedure for elderly patients could even be advantageous as we identified a trend towards benefits for the nESPC such as a reduction of CIT in the first portion or the HC and dialysis vintage as well as a better 5-year graft and comparable patient survival.

   Graft function, graft and patient survival are the main outcome parameters in KTx. They might be influenced by different factors including recipients' and donors' age, CIT, HLA mismatch and dialysis vintage [26, 27]. A major concern of the ESP was, therefore, that elderly recipients not only have to cope with the effects of their own age, but also experience disadvantages from the elderly graft as recipients' and donors' age has been shown to be a risk factor for death, perioperative and peri-transplantation complications [28–34]. Heldal et al. more specifically identified donors' age ≥60 years as a predictor for increased mortality and graft loss in the elderly [35]. Results from the UK, due to the different procurement policy only partly comparable to Germany, indicate that donor age is a risk factor for graft loss as well [11]. Dahmen et al. also identified donor's age as a major factor influencing future graft function [36]. The assumption that elderly grafts might negatively impact outcomes after KTx is therefore understandable. However, neither Neri nor Hwang identified an influence of donor age on graft function [6, 29]. In the data presented, both nESPC recipients and donors were significantly older. However, the older age of nESPC recipients and donors did not seem to influence the graft function, the graft or patient survival negatively.

   The elderly, especially, are more susceptible to damages by long CIT and its impact on graft function [29, 32, 37–39]. Hence, the ESP aimed for reduced CIT [10, 13]. However, reduction

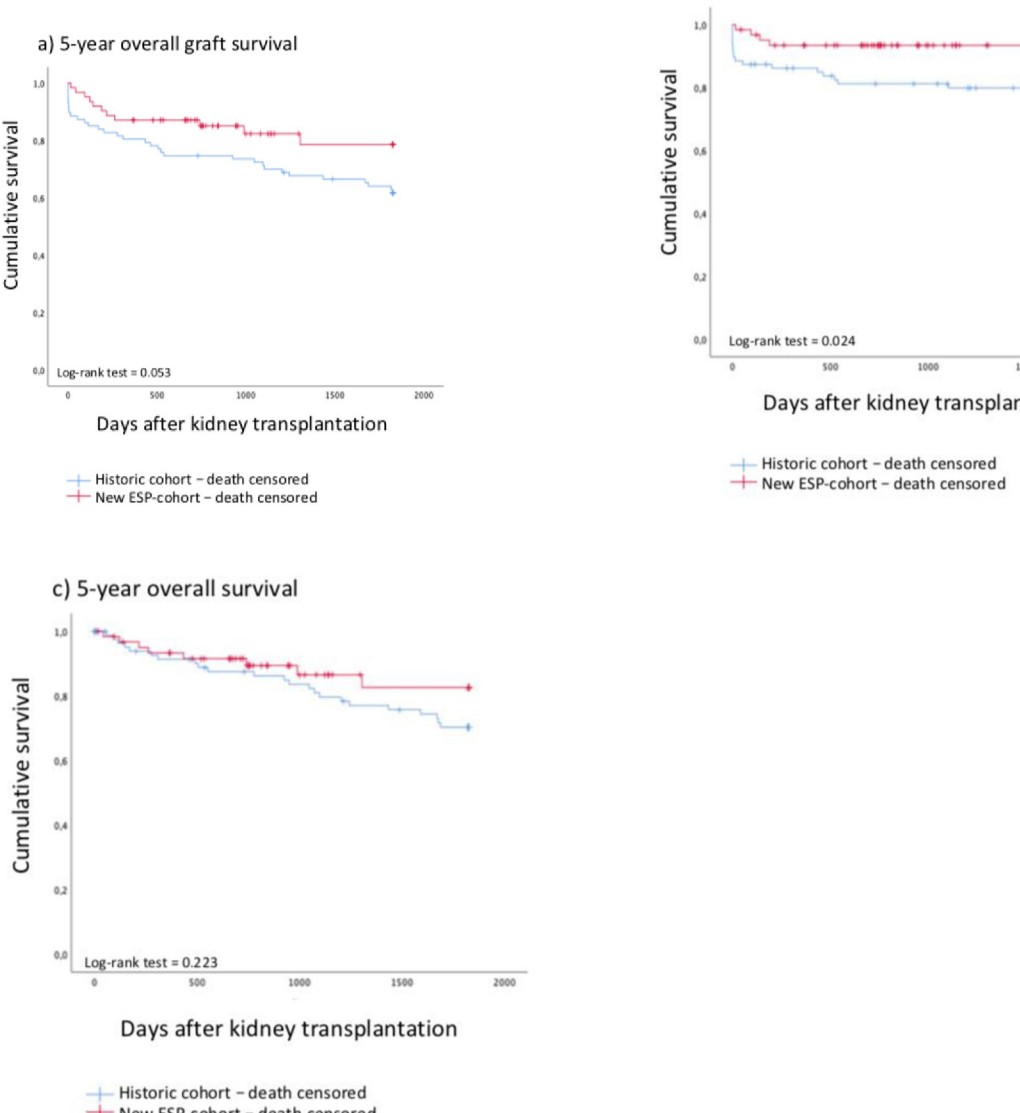

**Fig 2.** a) 5-year graft survival, b) 5-year death-censored graft survival and c) 5- patient survival stratified by cohorts. (blue HC, red nESPC). Kaplan-Meier-survival curves and log-rank test were used to compare survival. ESP = European Senior Program, y = year.

of CIT is often realized at the expense of a higher HLA mismatch which in turn affects graft function and survival [14]. Nonetheless, shorter CIT does not always fully compensate for the effect of HLA mismatches [18]. Despite a significant reduction of CIT after introduction of the nESP criteria, neither HLA mismatch, DGF nor PF significantly increased in the nESPC. However, the change in the allocation policy in 2005 had already led to a shortening of the CIT of the HC, resulting in comparable CIT in both cohorts after 2005. Moreover, the reduced CIT after 2005 might not only be attributable to the local allocation policy of the ESP but also results from better organization and transportation and an increased awareness of the relevance of CIT. Nonetheless, a reduced CIT is associated with a better outcome and may also have influenced the graft outcome in the nESPC.

**Table 5. Cox regression model for 5-year graft survival.**

| Cox regression model for 5-year graft survival | | | | |
|---|---|---|---|---|
| Parameters | Univariate | | Multivariable | |
| | HR (95% CI) | p-value | HR (95% CI) | p-value |
| HC vs nESPC | 2.094 (0.831–5.278) | 0.117 | | |
| Recipient age (years) | 0.953 (0.866–1.050) | 0.331 | | |
| Recipient sex (male vs female) | 1.229 (0.702–2.150) | 0.471 | | |
| Recipient BMI (>25 kg/m$^2$) | 2.304 (1.071–4.958) | **0.033** | 1.813 (1.175–2.799) | **0.012** |
| Time of RRT (months) | 1.012 (0.991–1.034) | 0.274 | | |
| Arterial hypertension recipient | 0.759 (0.368–1.564) | 0.455 | | |
| Diabetes recipient | 1.722 (0.846–3.504) | 0.134 | | |
| Hyperlipidemia recipient | 0.877 (0.423–1.820) | 0.725 | | |
| Arteriosclerosis recipient | 1.972 (1.059–3.701) | **0.035** | 1.371 (0.886–2.121) | 0.929 |
| Cold ischemia time (hours) | 0.930 (0.860–1.005) | 0.066 | | |
| Sum of HLA mismatches | 0.738 (0.562–0.970) | **0.029** | 0.968 (0.811–1.155) | 0.449 |
| Operation time (min) | 1.000 (1.000–1.000) | **0.018** | 1.000 (1.000–1.000) | 0.455 |
| Warm ischemia time (min) | 0.985 (0.961–1.010) | 0.235 | | |
| Intraoperative complications (yes vs no) | 1.615 (0.638–4.091) | 0.312 | | |
| Postoperative complications (yes vs no) | 1.357 (0.764–2.411) | 0.297 | | |
| Length of hospital stay (d) | 0.991 (0.974–1.009) | 0.320 | | |
| Donor age (years) | 1.011 (0.955–1.071) | 0.699 | | |
| Donor gender (male vs female) | 1.402 (0.743–2.645) | 0.297 | | |
| Donor BMI (kg/m$^2$) | 1.046 (0.966–1.134) | 0.270 | | |
| Donor creatinine pre-procurement (mg/dl) | 0.894 (0.442–1.809) | 0.756 | | |
| Arteriell Hypertension Donor | 1.314 (0.671–2.576) | 0.426 | | |
| Diabetes Donor | 1.168 (0.528–2.582) | 0.701 | | |
| Arteriosclerosis Donor | 0.748 (0.385–1.453) | 0.391 | | |
| Hyperlipidemia Donor | 0.614 (0.211–1.788) | 0.371 | | |
| Length of Donor on Intensive Care Unit (d) | 1.090 (1.023–1.161) | **0.007** | 1.046 (0.998–1.096) | 0.221 |
| PF (yes vs no) | 0.075 (0.018–0.308) | **< 0.001** | 1.400 (0.917–2.135) | 0.870 |
| Biopsy-proven Rejection (yes vs no) | 0.826 (0.512–1.335) | 0.435 | | |
| Intraoperative complications (yes vs no) | 1.235 (0.662–2.304) | 0.507 | | |
| Surgical complications (yes vs no) | 0.844 (0.569–1.252) | 0.401 | | |
| Non-surgical complications (yes vs no) | 0.716 (0.486–1.057) | 0.093 | | |

HR = hazard ratios, CI = 95% confidence interval. HC = historic cohort, ESP = European Senior Program, BMI = body mass index, HLA = human leukocyte antigen, PF = primary function.

HLA mismatch might be one reason for rejection [14, 40]. HLA A and especially HLA DR mismatches correlated with a higher incidence of rejection and graft loss [14, 16, 35, 41]. A low HLA mismatch should therefore be aimed for reduced numbers of rejection and their influence on graft survival [16, 41]. Because of increased DR mismatches in some ESPC resulting in increased numbers of rejection, Dreyer et al. advocated the extended "Eurotransplant Senior DR-compatible Program", favoring the maintenance of local allocation but not on the expenses of DR-compatibility [41]. However, emphasizing local allocation and short CIT within the nESP neither overall HLA mismatch increased nor significantly influenced rejection or infections. Even though DR mismatches did not increase in the nESPC at MTC, Dreyer et al.'s suggestions need to be considered as better DR matching can potentially further improve the outcome after KTx [14]. It should be noted that only one patient in the nESPC

**Table 6. Cox regression model for 5-year death-censored graft survival.**

| Cox regression model for 5-year death-censored graft survival | | | | |
| --- | --- | --- | --- | --- |
| Parameters | Univariate | | Multivariable | |
| | HR (95% CI) | p-value | HR (95% CI) | p-value |
| HC vs nESPC | 2.575 (1.016–6.527) | **0.046** | 1.210 (0.822–1.780) | 0.276 |
| Recipient age (years) | 0.937 (0.848–1.035) | 0.197 | | |
| Recipient sex (male vs female) | 1.739 (0.980–3.085) | 0.058 | | |
| Recipient BMI (kg/m$^2$) | 2.199 (0.994–4.862) | 0.052 | | |
| Time on dialysis (months) | 1.001 (0.989–1.034) | 0.342 | | |
| Arterial hypertension recipient | 0.647 (0.315–1.330) | 0.236 | | |
| Diabetes recipient | 2.300 (1.119–4.731) | **0.024** | 1.514 (0.983–2.331) | 0.403 |
| Hyperlipidemia recipient | 0.790 (0.372–1.681) | 0.514 | | |
| Arteriosclerosis recipient | 2.389 (1.246–4.580) | **0.009** | 1.318 (0.889–1.956) | 0.257 |
| Cold ischemia time (hours) | 0.947 (0.878–1.021) | 0.155 | | |
| Sum of HLA mismatches | 0.744 (0.567–0.975) | **0.032** | 0.925 (0.795–1.078) | 0.059 |
| Operation time (min) | 1.000 (1.000–1.000) | 0.097 | | |
| Warm ischemia time (min) | 0.982 (0.957–1.008) | 0.176 | | |
| Intraoperative complications (yes vs no) | 1.943 (0.730–5.172) | 0.184 | | |
| Postoperative complications (yes vs no) | 1.227 (0.698–2.159) | 0.478 | | |
| Length of hospital stay (days) | 0.999 (0.980–1.018) | 0.918 | | |
| Donor age (years) | 1.011 (0.953–1.072) | 0.726 | | |
| Donor gender (male vs female) | 1.202 (0.611–2.365) | 0.595 | | |
| Donor BMI (kg/m$^2$) | 1.050 (0.967–1.139) | 0.245 | | |
| Donor creatinine pre-procurement (mg/dl) | 2.575 (1.016–6.527) | 0.372 | | |
| Arteriell Hypertension Donor | 1.178 (0.599–2.317) | 0.643 | | |
| Diabetes Donor | 0.882 (0.377–2.065) | 0.773 | | |
| Hyperlipidemia Donor | 0.763 (0.260–2.233) | 0.621 | | |
| Arteriosclerosis Donor | 0.666 (0.347–1.277) | 0.221 | | |
| Length of Stay on Intensive Care Unit (donor) (d) | 1.062 (0.997–1.132) | 0.062 | | |
| PF (yes vs no) | 0.098 (0.022–0.437) | **0.002** | 1.596 (1.081–2.356) | 0.622 |
| Biopsy-proven rejection (yes vs no) | 0.695 (0.424–1.139) | 0.149 | | |
| Intraoperative complications (yes vs no) | 0.536 (0.240–1.195) | 0.127 | | |
| Surgical complications (yes vs no) | 0.800 (0.436–1.469) | 0.472 | | |
| Non-surgical complications (yes vs no) | 0.728 (0.406–1.304) | 0.286 | | |

HR = hazard ratios, CI = 95% confidence interval. HC = historic cohort, ESP = European Senior Program, BMI = body mass index, HLA = human leukocyte antigen, PF = primary function.

participated in the Eurotransplant Senior DR compatible Program, thus a potential bias posed by different HLA-DR matching can be excluded.

Graft function and survival are influenced by the dialysis vintage and the associated comorbidities [4, 19, 29, 30, 32, 33, 39]. Quick transplantation of an ECD graft and shorter dialysis vintage seem to outweigh the benefits of a younger graft despite a longer dialysis vintage [10]. The ESP hence aimed for a short dialysis vintage for elderly patients [13]. The listing of younger and elderly patients on a common waiting list probably leads to a longer waiting time for all patients with some patients dying awaiting KTx and with a competition between the age groups [3, 7]. In an analysis from our center comparing patients on the waiting list to patients receiving a graft with acute kidney injury of the donor prior to transplantation, we identified a 5-year survival of waiting list candidates of 41% only [25]. Between 2010 and 2014, the

**Table 7. Cox regression model for 5-year patient survival.**

| Cox regression model for 5-year patient survival | | | | |
|---|---|---|---|---|
| **Parameters** | **Univariate** | | **Multivariable** | |
| | **HR (95% CI)** | **p-value** | **HR(95% CI)** | **p-value** |
| HC vs nESPC | 1.829 (0.734–4.557) | 0.195 | | |
| Recipient age (years) | 0.955 (0.870–1.049) | 0.337 | | |
| Recipient sex (male vs female) | 2.100 (1.144–3.855) | **0.017** | 0.971 (0.634–1.489) | 0.422 |
| Recipient BMI (<25 kg/m$^2$) | 2.225 (0.984–5.032) | 0.055 | | |
| Time on dialysis (months) | 1.012 (0.990–1.035) | 0.287 | | |
| Arterial hypertension recipient | 0.617 (0.289–1.319) | 0.213 | | |
| Diabetes recipient | 2.671 (1.311–5.442) | **0.007** | 1.819 (1.132–2.921) | 0.090 |
| Hyperlipidemia recipient | 0.989 (0.476–2.503) | 0.976 | | |
| Arteriosclerosis recipient | 3.152 (1.620–6.132) | **0.001** | 1.379 (0.899–2.114) | 0.958 |
| Cold ischemia time (hours) | 0.911 (0.840–0.988) | **0.025** | 0.954 (0.910–1.000) | 0.272 |
| Sum of HLA mismatches | 0.830 (0.636–1.085) | 0.173 | | |
| Operation time (min) | 1.000 (1.000–1.000) | **0.010** | 1.000 (1.000–1.000) | 0.268 |
| Warm ischemia time (min) | 0.987 (0.961–1.013) | 0.310 | | |
| Intraoperative complications (yes vs no) | 2.212 (0.823–5.945) | 0.116 | | |
| Postoperative complications (yes vs no) | 1.134 (0.652–1.972) | 0.656 | | |
| Length of hospital stay (days) | 1.005 (0.986–1.025) | 0.606 | | |
| Donor age (years) | 0.976 (0.920–1.036) | 0.432 | | |
| Donor gender (male vs female) | 1.113 (0.547–2.266) | 0.767 | | |
| Donor BMI (kg/m$^2$) | 0.988 (0.912–1.069) | 0.761 | | |
| Donor creatinine pre-procurement (mg/dl) | 1.118 (0.533–2.345) | 0.769 | | |
| Arteriell Hypertension Donor | 1.310 (0.689–2.489) | 0.410 | | |
| Diabetes Donor | 1.404 (0.635–3.106) | 0.402 | | |
| Hyperlipidemia Donor | 2.157 (0.795–5.851) | 0.131 | | |
| Arteriosclerosis Donor | 0.793 (0.405–1.553) | 0.499 | | |
| Length of Donor on Intensive Care Unit (d) | 1.083 (1.017–1.153) | **0.013** | 1.053 (1.000–1.109) | 0.351 |
| PF (yes vs no) | 0.272 (0.065–1.143) | 0.076 | | |
| Biopsy-proven rejection (yes vs no) | 0.707 (0.419–1.193) | 0.194 | | |
| Intraoperative complications (yes vs no) | 0.572 (0.297–1.102) | 0.095 | | |
| Postoperative surgical complications (yes vs no) | 0.954 (0.634–1.436) | 0.822 | | |
| Non-surgical complications (yes vs no) | 0.510 (0.338–0.769) | **<0.001** | 0.198 (0.015–0.272) | **0.029** |

HR = hazard ratios, CI = 95% confidence interval. HC = historic cohort, ESP = European Senior Program, BMI = body mass index, HLA = human leukocyte antigen, PF = primary function.

inclusion period of the nESPC in this paper, 21.7% of the patients who died on the waiting list were part of the ESP. During a shortage of organs, the waiting time usually increases. The significantly shorter dialysis vintage in our nESPC is against the general trend and may itself highly influence graft outcome [10]. However, Frei et al. reported a similar tendency [18]. It is noteworthy that since 2010, the number of actively listed patients decreased in Germany and in the Eurotransplant region. This might be attributable to the extended donor pool [10].

Evaluation of the results presented must consider the limitations of a retrospective data base analysis including its non-avoidable selection bias and limited statistical power due to the relatively small numbers of patients included. Unfortunately, we have not found a better way to compare both cohorts. We have included the most common comorbidities in CKD-patients in order to analyze their influence on graft outcome. However, other comorbidities and other

unknown confounders not included in our analysis may also influence outcome after RRT and RTx. Correlating graft function with creatinine clearance calculated according to Modification of Diet in Renal Disease Study Group (MDRD) is a methological limitation as creatinine levels depend on individual parameters such as dietary intake or muscle masses among other factors. Further, inclusion periods were quite long with changes occurring concerning procurement procedures such as the change of preservation solution and surgical procedures such as splinting of the ureterocystostomy, the intra- and postoperative complications, allocation criteria and improvement of immunosuppressive therapies. These surgical as well as medical changes may influence the outcome of transplantation. Yet, more intra- and postoperative complications were registered in the nESPC still resulting in comparable longterm graft function. The increase in bone marrow toxicity may be due to the change in immunosuppressive therapy. Additionally, complete 5-year follow-up was relatively small with some gaps in the inclusion period resulting in limited statistical power. However, 3 year overall patient and graft survival did not differ noticeably. Yet, 3-year death censored graft survival was noticeably better in the new cohort (p = 0.037 vs HC).

To our knowledge, this is the first study comparing the aforementioned changes in the allocation procedure in a German cohort showing no disadvantages for ESP recipients with a single listing with regard to graft and patient survival even though recipients and donors in the nESPC were significantly older. The new ESP criteria led to a significant reduction of waiting time and CIT in our center. This was not achieved at the expense of HLA mismatch, resulting in comparable numbers of primary and overall graft function and rejections. ESP-recipients after single listing therefore do not seem to be penalized by the single listing.

## Author Contributions

**Conceptualization:** Anne-Sophie Mehdorn, Stefan Reuter, Barbara Suwelack, Felix Becker, Norbert Senninger, Daniel Palmes, Ralf Bahde.

**Data curation:** Anne-Sophie Mehdorn, Katharina Schütte-Nütgen, Felix Becker, Thomas Vogel, Ralf Bahde.

**Formal analysis:** Anne-Sophie Mehdorn, Katharina Schütte-Nütgen, Ralf Bahde.

**Funding acquisition:** Stefan Reuter.

**Project administration:** Anne-Sophie Mehdorn, Stefan Reuter.

**Supervision:** Stefan Reuter, Barbara Suwelack, Norbert Senninger, Daniel Palmes, Thomas Vogel, Ralf Bahde.

**Validation:** Stefan Reuter, Felix Becker, Ralf Bahde.

**Writing – original draft:** Anne-Sophie Mehdorn.

**Writing – review & editing:** Anne-Sophie Mehdorn, Stefan Reuter, Barbara Suwelack, Katharina Schütte-Nütgen, Felix Becker, Norbert Senninger, Daniel Palmes, Thomas Vogel, Ralf Bahde.

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
