## [Decision Letter · Decision Letter 0]

17 Mar 2020

PONE-D-20-01629

Comparison of Kidney Allograft Survival in the Eurotransplant Senior Program after Changing the Allocation Criteria in 2010 – A Single Center Experience

PLOS ONE

Dear Prof. Reuter,

Thank you for submitting your manuscript to PLOS ONE. After careful consideration, we feel that it has merit but does not fully meet PLOS ONE’s publication criteria as it currently stands. Therefore, we invite you to submit a revised version of the manuscript that addresses the points raised during the review process.

ACADEMIC EDITOR: 

Overall the manuscript is of interest to the renal transplant community. However, it would benefit from providing more detailed both methodology and results description. For that reason I would ask you to provide the following information:

Please clarify if 89 patients in the historic cohort are all those who got an ESP kidney. I so, please provide data on the patients who received a kidney on the standard wait list from the historic cohortThe ESPD program actually started in 2010 randomly assigning kidney from elderly donor as pairs with or without matching for HLA-DR. Please state clearly whether the 62 newESP recipients were actually part of this ESDP initiative? If yes, please analyze the potential reasons for the better long-term eGFR including matching and rejection episodes and/or de novo HLA-DR/DQ antibody formationPlease provide the proportions of patients in the first cohort that received CsA and basiliximabPlease provide information on the criteria for including variables in the multivariable modelPlease provide rationale for the choice of analyzed comorbidities

The Authors should also acknowledge all limitations of the study pointed out by the Reviewers.

There are also other minor issues, which are described in detail by the Reviewers.

We would appreciate receiving your revised manuscript by May 01 2020 11:59PM. To enhance the reproducibility of your results, we recommend that if applicable you deposit your laboratory protocols in protocols.io, where a protocol can be assigned its own identifier (DOI) such that it can be cited independently in the future. For instructions see: http://journals.plos.org/plosone/s/submission-guidelines#loc-laboratory-protocols

We look forward to receiving your revised manuscript.

Kind regards,

Justyna Gołębiewska

Academic Editor

PLOS ONE

Journal Requirements:

2. We note that your study involved tissue/organ transplantation. Please provide the following information regarding tissue/organ donors for transplantation cases analyzed in your study. 1. Please provide the source(s) of the transplanted tissue/organs used in the study, including the institution name and a non-identifying description of the donor(s). 2. Please state in your response letter and ethics statement whether the transplant cases for this study involved any vulnerable populations; for example, tissue/organs from prisoners, subjects with reduced mental capacity due to illness or age, or minors. - If a vulnerable population was used, please describe the population, justify the decision to use tissue/organ donations from this group, and clearly describe what measures were taken in the informed consent procedure to assure protection of the vulnerable group and avoid coercion. - If a vulnerable population was not used, please state in your ethics statement, “None of the transplant donors was from a vulnerable population and all donors or next of kin provided written informed consent that was freely given.” 3. In the Methods, please provide detailed information about the procedure by which informed consent was obtained from organ/tissue donors or their next of kin. In addition, please provide a blank example of the form used to obtain consent from donors, and an English translation if the original is in a different language. 4. Please indicate whether the donors were previously registered as organ donors. If tissues/organs were obtained from deceased donors or cadavers, please provide details as to the donors’ cause(s) of death. 5. Please provide the participant recruitment dates and the period during which transplant procedures were done (as month and year). 6. Please discuss whether medical costs were covered or other cash payments were provided to the family of the donor. If so, please specify the value of this support (in local currency and equivalent to U.S. dollars).

4. Please include a new copy of Table 1-7 in your manuscript; the current table is difficult to read. Please follow the link for more information: http://blogs.PLOS.org/everyone/2011/05/10/how-to-check-your-manuscript-image-quality-in-editorial-manager/

5. We note you have included a table to which you do not refer in the text of your manuscript. Please ensure that you refer to Table 1-7 in your text; if accepted, production will need this reference to link the reader to the Table.

6. Please include your tables as part of your main manuscript and remove the individual files. Please note that supplementary tables (should remain/ be uploaded) as separate "supporting information" files

Reviewers' comments:

Reviewer's Responses to Questions

**Comments to the Author**

1. Is the manuscript technically sound, and do the data support the conclusions?

Reviewer #1: No

Reviewer #2: Yes

Reviewer #3: Yes

Reviewer #4: Partly

2. Has the statistical analysis been performed appropriately and rigorously? 

Reviewer #1: No

Reviewer #2: Yes

Reviewer #3: Yes

Reviewer #4: Yes

3. Have the authors made all data underlying the findings in their manuscript fully available?

Reviewer #1: Yes

Reviewer #2: No

Reviewer #3: Yes

Reviewer #4: Yes

4. Is the manuscript presented in an intelligible fashion and written in standard English?

Reviewer #1: Yes

Reviewer #2: Yes

Reviewer #3: Yes

Reviewer #4: Yes

5. Review Comments to the Author

Reviewer #1: Thank you for the invitation to review the manuscript by AS Medhorn et al entiteled “Comparison of Kidney Allograft Survival in the Eurotransplant Senior Program after Changing the Allocation Criteria in 2010 – A Single Center Experience” for publication in PLOS One.

As the number of patients with ESRD, especially those older than 65 years, is increasing without a corresponding increase in available donor organs, the waiting lists and waiting time increase and make it important to develop allocation programs designed to increase the organ pool. The European Senior Program (ESP) was designed to allocate grafts from donors older than 65 years to non-immunized older (> 65 years) patients listed for transplantation. The allocation was performed without taking HLA matching into account thereby focusing on short cold ischemia time (CIT) and shorten the waiting time/dialysis time for older patients since they did not have to compete with the younger patients to get an ESP organ. In the beginning, patients were listed both on the standard waiting list and the ESP list, but after 2010 the rules were changed and patient had to choose one of the lists. The aim of the study was to determine whether this change puts elderly patients at a disadvantage compared to a cohort listed on both waiting lists.

It is important to evaluate any changes in practice that may influence the results either positively or negatively. However, I believe this study has a number of major limitations that make it difficult to draw any conclusions. I will briefly describe these limitations.

1. Cohort differences

a. They are from different time periods. Registry data have shown that survival after transplantation has improved with time and consequently it is difficult to know if the observed results are due to change in practice or a result of different time cohorts

b. There are major differences in the surgical procedure regarding the ureterocystostomy between the cohorts. It is also a remarkable difference in intraoperative complications between the cohort, a difference that cannot be expected to be caused by change in the practice of listing.

c. The immunosuppressive regimen also differs between the cohorts, Cyclosporin was used between 1999 and 2002 and tacrolimus in the rest of the study period. Basilkiximab was introduced in 2006 and consequently the historical cohort consists of a mix of imunosuppresive regimens. It is not described the proportions of patients in the first cohort that received CsA and basiliximab.

2. End points

a. Graft function described by serum creatinine and eGFR calculated according to Cockcroft-Gault´s equation was chosen as primary end point. These are not very accurate, especially s-creatinine differs markedly between patients according to muscle mass. In addition, the Cockcroft-Gault equation does not calculate the eGFR but creatinine clearance.

b. Graft and patient survival were chosen as secondary end-points. I believe that they should rather be primary end points, but obviously the observation period is to short to evaluate the 5-year survival as only 13% of patients were followed for five years.

3. Survival statistics

a. The description of the cox regression analysis can´t be correct. I suppose that the authors intended to state that variables with possible association with five-year patient and graft survival were evaluated in univariable cox regression models and that noticeable values were included in multivariable Cox models (not linear). The authors have not described the criteria for including variables in the multivariable model and in the tables describing the uni- and multivariable hazard ratios, only one variable is presented in the multivariable model. All variables included in the multivariable model should be included in the table with HR and 95% CI

b. The log-rank test is a test comparing the survival curves and not the actual 5-year survival. The p-values should consequently be included in the KM-plots and not in the comparison of 5-year survival proportions. Since only a small proportion of patients were followed for five years the 5-year survival estimates are associated with a major uncertainty.

4. Comorbidities

a. It is not described which comorbidities that were included and which were not. Table 1 describes the proportions of hypertension, diabetes, arteriosclerosis and hyperlipidemia. Why were these chosen? Why not ischemic heart disease, heart failure or cerebrovascular disease. Several comorbidity scores are developed to describe comorbidity. One of these should be included. Since the study is a single center study including a relatively low number of patients, it should be possible to calculate a comorbidity index for all patients based on hospital records.

5. Statistical strength

a. A total of 151 patients were included, 62 in the first cohort and 89 in the second cohort. In several analysis this is obviously too few to draw any firm conclusions. For instance, the proportion of intraoperative complications is five-fold higher in the. HC compared to the nESP cohort (3.1 % vs. 15.7 %). The difference is however not significant (p=0.283)

Reviewer #2: The authors report their single center experience transplanting elderly recipients (n=151) with older donor kidney (“old-for-old”) according to either a restricted waiting list (newESP; n=62) or the historic option within Eurotransplant allowing placement on both the regular (ETKAS) as well as the ESP waiting list (n=89).

The primary end-point (eGFR by Cockcroft-Gault) was comparable in the two allocation groups. Recipients of the newESP kidneys had lower creatinine levels at 18 and 60 months post-transplant. The eGFR difference reached significance at the 5-year point.

There were significant differences in both the recipient and donor characteristics between newESP and historic controls. Multivariable Cox regression showed no significant differences in death-censored (except diabetics), patient and (overall) graft survival (except BMI). Results are plotted using (univariable) Kaplan-Meier curves.

Overall the manuscript is of interest, well written and adequately referenced. The only significant and clinically relevant difference was observed in long-term kidney graft function in favor of the newESP allocation principle. The authors adequately discuss potential differences including a potential role for clinical immunosuppression and prospective matching for HLA class-II antigens.

The ESPD program actually started in 2010 randomly assigning kidney from elderly donor as paires with or without matching for HLA-DR.

1) The authors should clearly state whether their 62 newESP recipients were actually part of this ESDP initiative?

2) If yes, the better long-term function could be explained by less acute rejection epsides or less steroid resistant episodes and/or less de novo HLa-DR/DQ antibody formation

Reviewer #3: The authors presented the results of kidney transplantation in elderly recipients using a new allocation strategy. Although the results are excellent, the authors should better clarify how these organs were allocated: in other words, does this cohort include consecutive patients or patients were selected. Could this new allocation strategy penalize the patients who are not included in this group. Moreover, the authors should better detail in the discussion that the two group were not homogeneous, and that there is a great disparity in terms of cold ischemia time and dialysis vintage, which are well known factors strongly influencing the outcome of kidney transplantation.

Reviewer #4: Could you detail how patients now decide which waiting list to go on?

I have a concern about the comparison groups however....it is not exactly clear what you did. In lines 74- 76 you say you seek to determine whether single listing puts older patients at a disadvantage now compared to when they were wait listed on both lists but unless I have misunderstood you don't seem to answer that question because you say in your final analysis you only include data from the historic cohort who received an ESP kidney (Line 85 - you say only patients who received an esp kidney were included.) Does this mean that your 89 patients in the historic cohort are all those who got an ESP kidney? How many other historic cohort patients got a Tx from the standard list?

If I have understood the above correctly then you need to provide data on the patients who received a kidney on the standard wait list from the historic cohort. otherwise the comparison is therefore misleading. How many patients in either cohort on either list died whilst on the list?

a more meaningful comparison would be either the inclusion of all data from the historic cohort ie those who received a transplant on the standard ETKAS as well as the ESP list? Would be of interest to see how they fared compared to those in this cohort who received a Tx on the ESP list and in fact potentially a more useful and meaningful comparison. how long did they wait etc? And/ or a comparison in the new cohort of those who select to go onto the standard wait list v those who elect for the ESP wait list including the demographics. Is there inherent selection bias in those who select one or other list(s)?

In terms of results:

the new cohort compared to the historic cohort are older and heavier, have been on dialysis for a shorter time period, have more hypertension and diabetes but less arteriosclerosis (presumably a factor of shorter dialysis vintage), There are more post op surgical and non surgical complications. Non significant trends towards less DGF, better graft survival and higher levels of rejection.

The number for 3 years death censored graft survival is odd in the historic cohort - 33.9%. can you explain?

The cox regression analysis - you do univariate and not multivariate analysis presumably due to the small numbers.

You don't offer an explanation then of why graft survival is better long term? You allude to this in the discussion - ie different surgical techniques, improved immunosuppression regimens, altered preservation/retrieval which takes me back to my first point a better comparison not be of all patients in historic cohort and all patients over 65 on either list in the new cohort?

Figures and tables.

For the Kaplan Meier graphs, I would include the p values and the raw numbers on the curves so it is immediately obvious. Creatinine not keratinin. Similarly need to provide p values and what Asterix represents on the eGFR and creatinine bar charts either in the legend or on figure itself.

Minor points and language/grammatical issues:

For the values consistency so quote either 1 or 2 decimal places not a mixture

Line 22-24 This sentence in the abstract needs reworded - it does not make sense as it stands

Line 52 would change offer to supply and after : should be a not A

Line 83 An not a

Line 101 mycophenolate

Line 104 co-trimoxazole

Line 121 rejection rather than rejections

Line 124 and 125 as not in

Line 126 change noticeable to significant or could use notable but noticeable is incorrect

Line 128 hazard ratios

Line 129 Noticeable to signifcant or as for line 126

Line 166 primary functions - better to say towards less DGF or tended to be more likely to achieve primary function.

Line 179 - rejection should not be pleural

Line 180 - Acute rejection episodes were non significantly increased in the nESPC

Line 180 add the presence of diabetes

Line 191 lymphocele singular

Line 222 - The elderly, especially,

Line 225 shorter CIT does not always fully compensate for the effect

Line 233 and 234 rejection singular

6. PLOS authors have the option to publish the peer review history of their article (what does this mean?). If published, this will include your full peer review and any attached files.

Reviewer #1: No

Reviewer #2: No

Reviewer #3: No

Reviewer #4: No

---

## [Author Response · Author response to Decision Letter 0]

14 May 2020

Please see the attached respnose letter as it contains new figures/tables, whicht we are unabel to include in this text field.

Academic editor

1. Please clarify if 89 patients in the historic cohort are all those who got an ESP kidney. I so, please provide data on the patients who received a kidney on the standard wait list from the historic cohort.

We thank the academic editor for this question and appologize not making this important point clearer in our initial manuscript. Precisely, all 89 patients in the historic cohort received an ESP kidney (line 87). 

To further address this issue, we have now included a more precise description in the results section, subsection “study population” (line 146). 

Old: 

Between January 1999 and May 2007, 89 elderly patients had transplants after being listed on both waiting lists (HC). 

New:

Between January 1999 and May 2007, 89 elderly patients had transplants after being listed on both waiting lists (HC). Yet, only recipients of an ESP-kidney were included in the final analysis.

2. The ESPD program actually started in 2010 randomly assigning kidney from elderly donor as pairs with or without matching for HLA-DR. Please state clearly whether the 62 newESP recipients were actually part of this ESDP initiative? If yes, please analyze the potential reasons for the better long-term eGFR including matching and rejection episodes and/or de novo HLA-DR/DQ antibody formation

We thank the academic editor for this excellent question. Indeed, our center participated in the ESDP initiative. However, we only randomized one patient during the recruitment periode. Based on this small numner, we did not take this into account when analysing our dataset and did not mention it in the initial manuscript. Since we completely agree wit the editor that this could a potential bias in our results and should be clarified for the readership, we have now included the following statement in our revised manuscript.

Old:

Even though DR-mismatches did not increase in the nESPC at MTC, Dreyer et al.’s suggestions need to be considered as better DR-matching potentially further improves outcome after KTx [1]. 

New:

Even though DR-mismatches did not increase in the nESPC at MTC, Dreyer et al.’s suggestions need to be considered as better DR-matching potentially further improves outcome after KTx [1]. It should be noted that only one patient in the nESPC participated in the Eurotransplant Senior DR compatible Program, thus a potential bias posed by different HLA-DR matching can be excluded.

3. Please provide the proportions of patients in the first cohort that received CsA and basiliximab.

We thank the editor for raising this important point regarding the induction and maintenance immunospressiv regiems in our cohort. Accordingly, we have added the following statement in our reviewed manuscript:

Old:

Seven nESPC recipients and 12 HC-patients had previously undergone KTx before receiving an ESP kidney. 

New: 

Seven nESPC recipients and 12 HC-patients had previously undergone KTx before receiving an ESP kidney. A total of 68 HC-patients received an induction therapy with a monoclonal IL-2 receptor antibody (basiliximab) and the most common (used for 47 patients) immunosuppressive maintenance regime was tripel therapy consisting of cyclosporine, mycophenolate mofetile and steroids until 2003. Hereafter, maintenance regimes consisted of tacrolimus, mycophenolate mofetil and steroids. 

4. Please provide information on the criteria for including variables in the multivariable model

We thank the academic reviwer for bringing up this important point and applogize for not being more presice in the description of our statistical methods. For multivariable model building, we conducted a stepwise variable selection procedure. All covariates with a p-value less than 0.05 were included, while all variables with a p-value in the likelihood ratio test > 0.1 were excluded from further analysis.

To further clarify this method, we have now added the following statement to our material and methods section:

Old:

Hazard’s ratio, corresponding 95%-confidence intervals and p-values were reported. Significant values were further analyzed by multivariate linear regression.

New:

Hazard’s ratio, corresponding 95%-confidence intervals and p-values were reported. For multivariable model building, variables with a p-value in the likelihood ratio test > 0.1 were excluded and only variables with a p-value less than 0.05 were further analyzed by multivariate linear regression.

5. Please provide rationale for the choice of analyzed comorbidities

We thank the editor for this valuable comment regarding the analyzed comorbidities. We acknowledge that our initial manuscript lacked a more precise explanation regarding our selection for analyzed comorbidities. 

We have choosen arterial hypertension and diabetes mellitus since they are well described as major risk factors in patients with chronic kidney diseases, influencing outcome on dialysis as well as after transplantation (Li PKT, Garcia-Garcia G, Lui SF, Andreoli S, et al. Arch Argent Pediatr 2020). In addition, hyperlipidemia and adipositas are part of the metabolic syndrome, which influences outcome in patients with chronic kidney diseases and may cause arteriosclerosis. Arteriosclerosis may influence the surgical outcome and procedures (Hernández D, Alonso-Titos J, Armas-Padrón AM, Kidney Blood Press Res. 2020). 

To better address this issue, we have added the following statement:

New:

Recipient data included demographic data, underlying renal disease, most common comorbidities and time of dialysis vintage [2-4]. To further characterize RTx-recipients, comorbidities, including arterial hypertension, diabetes mellitus, hyperlipidemia and arteriosclerosis as well as BMI as a surrogate for overweight were included in the analysis. Arterial hypertension and diabetes mellitus are well described major risk factors in patients with chronic kidney diseases as are hyperlipidemia and overweight [2]. Arteriosclerosis may influence surgical outcome and was therefor included as well [3]. 

 

Journal requirements

We thank the editorial team for this most helpful advice and have adjusted the manuscript’s style accordingly. 

2. We note that your study involved tissue/organ transplantation. Please provide the following information regarding tissue/organ donors for transplantation cases analyzed in your study. 

Please provide the source(s) of the transplanted tissue/organs used in the study, including the institution name and a non-identifying description of the donor(s). 

All organs included in this study were procured from donors in the Eurotransplant region. Allocation and distribution were coordinated by Eurotransplant. Distribution in Germany was coordinated by the German Foundation for Organtransplantation (Deutsche Stiftung Organtransplantation, DSO). A description of all donors included can be found in Table. 2. 

Please state in your response letter and ethics statement whether the transplant cases for this study involved any vulnerable populations; for example, tissue/organs from prisoners, subjects with reduced mental capacity due to illness or age, or minors. - If a vulnerable population was used, please describe the population, justify the decision to use tissue/organ donations from this group, and clearly describe what measures were taken in the informed consent procedure to assure protection of the vulnerable group and avoid coercion. - If a vulnerable population was not used, please state in your ethics statement, “None of the transplant donors was from a vulnerable population and all donors or next of kin provided written informed consent that was freely given.” 

No organs from prisoners, subjects with reduced mental capacity due to illness or age or minors were included in the study. All donors included were >65 years, as this was one of the inclusion criteria. All donors included had given informed consent to the donation or their relatives had decided on their behalf acknowledging the potential will of the deceased. We have adjusted the sections accordingly.

New:

None of the transplant donors was from a vulnerable population and all donors next of kin provided written informed consent that was freely given. 

New:

No donations from prisoners, minor donors or people with reduced mental health were included. 

In the Methods, please provide detailed information about the procedure by which informed consent was obtained from organ/tissue donors or their next of kin. In addition, please provide a blank example of the form used to obtain consent from donors, and an English translation if the original is in a different language. 

New:

Informed consent for organ donation was obtained according to German organ donation law. Patients either decided during lifetime if they wanted to become organ donor. If patients were considered eligible for donation and had not given written consent prior to their death, close relatives decided on the patient behalf after the patient’s death acknowledging the potential will of the deceased. 

Please indicate whether the donors were previously registered as organ donors. If tissues/organs were obtained from deceased donors or cadavers, please provide details as to the donors’ cause(s) of death.

According to German transplantation law all donors were registred as donors prior to precurement. All donations were from heart beating, brain dead donors aged ≥65 years (line 107). Donors died because of intracranial bleeding (43.5% and 47.2%, respecitvely), subarachonidal bleeding (19.4% and 29.2%, respectively), cerebral infarction (22.6% and 10.1%, respectively), head trauma (9.7% and 13.5%, respectively) and hypoxic damage (3.2% and 0%, respectively) (see Table 2 for further details). 

Please provide the participant recruitment dates and the period during which transplant procedures were done (as month and year). 

We thank the editorial team for this important comment. At Muenster Transplant Center kindey transplantations have been performed since the 1970s. However, patients included in this puplication recieved kidney transplantations between October 2010 and September 2014 and January 1999 and May 2007. German transplantation law states in §3 section 3 sentence 1 that the treating physician must contact the transplantation center which should perform transplantation as soon as the eligible patient has given informed written consent in order to list the patient on the waiting list. Therfore, recruitment time can be either considered to start at time of dialysis or at the time when a patient is actively listed for kidney transplantation. However, we consider the later one as to vague as patients may start dialysis and only consider kidney transplantation later on, thus we included time of renal replacement therapy for better comparisson in our analysis, which represent the start of recruting time. 

Old:

However, only patients who received an ESP kidney were included in the final analysis. 

New:

However, only patients who received an ESP kidney were included in the final analysis. Recipient data included demographic data, underlying renal disease, most common comorbidities and time of dialysis vintage [2-4]. The dialysis vintage served as surrogate for the recruiting period.

Please discuss whether medical costs were covered or other cash payments were provided to the family of the donor. If so, please specify the value of this support (in local currency and equivalent to U.S. dollars).

Due to German transplant law medical costs for the procedure and further costs were covered by medical insurrance. The family of the donor did not receive any financial support or cash payments. 

3. PLOS requires an ORCID iD for the corresponding author in Editorial Manager on papers submitted after December 6th, 2016. 

We have included the corresponding author’s ORCID, which is as follows: 0000-0001-6345-6206

4. Please include a new copy of Table 1-7 in your manuscript; the current table is difficult to read. 

We thank the editorial board for the advice and have modified the tables accordingly. 

5. We note you have included a table to which you do not refer in the text of your manuscript. Please ensure that you refer to Table 1-7 in your text; if accepted, production will need this reference to link the reader to the Table.

We thank the editorial team for this hint and have now included all tables in the manuskript and marked the insertion accordingly. 

Tab 1 line 145, 

Tab 2 line 158

Tab 3 lines 166, 190, 192 and 200

Tab 4 lines 170 and 180

Tab 5 line 186

Tab 6 line 186

Tab 7 line 199, 

Fig 1 line 179, Fig 2 line 177. 

6. Please include your tables as part of your main manuscript and remove the individual files. Please note that supplementary tables (should remain/ be uploaded) as separate "supporting information" files

We thank the editorial team for this comment. All tables and figures have been adjusted and included at the end of main manuskript. 

 

Review Comments to the Author

Reviewer #1: 

(…) I believe this study has a number of major limitations that make it difficult to draw any conclusions. I will briefly describe these limitations.

Cohort differences

a. They are from different time periods. Registry data have shown that survival after transplantation has improved with time and consequently it is difficult to know if the observed results are due to change in practice or a result of different time cohorts

We thank the reviewer for this important comment and acknowledge this point, which we fully agree with. Unfortunatelly, we do not have any better way to compare the influence of the change in allocation policy. 

We also agree that surgical as well as medical changes have been made in that period of time, which could be a potential bias when interpretating our results. Since we fully concur with the reviwer that our cohorts encompass patients from different time periods and that this itself poses a potential bias, we included a statement in our reviced manuscript to better disclose this limitation of our study.

Old: 

Further, inclusion periods were quite long with changes occurring concerning procurement procedures such as the change of preservation solution and surgical procedures such as splinting of the ureterocystostomy, allocation criteria and improvement of immunosuppressive therapies. 

New:

Further, inclusion periods were quite long with changes occurring concerning procurement procedures such as the change of preservation solution and surgical procedures such as splinting of the ureterocystostomy, the intra- and postoperative complications, allocation criteria and improvement of immunosuppressive therapies. These surgical as well as medical changes may have influenced outcomes in the here presented dataset. 

b. There are major differences in the surgical procedure regarding the ureterocystostomy between the cohorts. It is also a remarkable difference in intraoperative complications between the cohort, a difference that cannot be expected to be caused by change in the practice of listing.

We thank the reviwer for this valuable comment and acknowledge the concern. We agree that there are mayor differences with regards to surgical procedures and complications between the two cohorts. We also agree that complications may influence outcome. Yet, more postoperative complications were registered in the nESPC still resulting in comparable longterm graft function. The increase in bone marrow toxicity may be due to the change in immunosuppressive therapy. However, we think that the reduced cold ischemia time results from different reasons, such as change in the allocation policy, but also better logistics. Reduced cold ischemia time itself is said to have a significant influence on outcome after transplantation. However, it is also possible that this is due to a change in analyzing the data. We have included this point in the limitations section of the discussion (line 308).

Old:

Further, inclusion periods were quite long with changes occurring concerning procurement procedures such as the change of preservation solution and surgical procedures such as splinting of the ureterocystostomy, allocation criteria and improvement of immunosuppressive therapies.

New:

Further, inclusion periods were quite long with changes occurring concerning procurement procedures such as the change of preservation solution and surgical procedures such as splinting of the ureterocystostomy, the intra- and postoperative complications, allocation criteria and improvement of immunosuppressive therapies. These surgical as well as medical changes may influence the outcome of transplantation. Yet, more postoperative complications were registered in the nESPC without still resulting in comparable longterm graft function. The increase in bone marrow toxicity may be due to the change in immunosuppressive therapy.

c. The immunosuppressive regimen also differs between the cohorts, Cyclosporin was used between 1999 and 2002 and tacrolimus in the rest of the study period. Basilkiximab was introduced in 2006 and consequently the historical cohort consists of a mix of imunosuppresive regimens. It is not described the proportions of patients in the first cohort that received CsA and basiliximab.

We thank the reviewer for this helpful remark. We added the numbers in the respective paragraph. 

Old:

Seven nESPC recipients and 12 HC-patients had previously undergone KTx before receiving an ESP kidney. 

New:

Seven nESPC recipients and 12 HC-patients had previously undergone KTx before receiving an ESP kidney. A total of 68 HC-patients received an induction therapy with a monoclonal IL-2 receptor antibody (basiliximab) and the most common (used for 47 patients) maintenance regime was trippel therapy consisting of cyclosporine, mycophenolate mofetile and steroids until 2003. Hereafter, maintenance regimes consisted of tacrolimus, mycophenolate mofetil and steroids. 

End points

a. Graft function described by serum creatinine and eGFR calculated according to Cockcroft-Gault´s equation was chosen as primary end point. These are not very accurate, especially s-creatinine differs markedly between patients according to muscle mass. In addition, the Cockcroft-Gault equation does not calculate the eGFR but creatinine clearance.

We thank the reviewer for this important point and have changed all respective paragraphs accordingly. We have chosen serum creatinine and eGFR as primary endpoint since we think this parameters provide reasonable overview regarding graft function. However, we are fully aware of the limitation of serum creatinin levels and false high or low creatinine levels depending on dietary intake and muscle mass. Yet, as far as we know there is no better way of calculating graft function. 

To better disclose this limitation, we have added the following paragraph:

Old:

The primary endpoint was graft function using serum creatinine and estimated glomerular filtration rate (eGFR) calculated according to Cockroft-Gault’s equation as surrogates.

New:

The primary endpoint was graft function using serum creatinine and estimated glomerular filtration rate (eGFR) calculated according Modification of Diet in Renal Disease Study Group (MDRD) as surrogates (22). Graft function was chosen as primary endpoint due to the small number of patients included. However, we are aware of the limitations of correlating graft function according to MDRD, as creatinine levels depend on dietary intake and muscle mass among other factors. 

Old: 

Evaluation of the results presented must consider the limitations of a retrospective data base analysis including its non-avoidable selection bias and limited statistical power due to the relatively small numbers of patients included. 

New:

Evaluation of the results presented must consider the limitations of a retrospective data base analysis including its non-avoidable selection bias and limited statistical power due to the relatively small numbers of patients included. Unfortunately, we have not found a better way to compare both cohorts. Correlating graft function with creatinine clearance calculated according to Modification of Diet in Renal Disease Study Group (MDRD) is a methological limitation as well as creatinine levels depend individual parameters such as dietary intake or, muscle mass among other factors.

b. Graft and patient survival were chosen as secondary end-points. I believe that they should rather be primary end points, but obviously the observation period is to short to evaluate the 5-year survival as only 13% of patients were followed for five years.

We thank the esteemed referee for this comment and acknowledge the concern. We fully agree with the concern. Unfortunatelly, this seems to be one of the limitations of a registry based retrospective study. In order to face this limitation, we also calculated 3-year survival for both cohorts (Table 4). 

Old: 

Aditionally, complete 5-year follow-up was relatively small with some gaps in the inclusion period.

New:

Aditionally, complete 5-year follow-up was relatively small with some gaps in the inclusion period resulting in limited statistical power. 

Old:

The 5-year overall follow-up was 13.4% (nESPC) and 12.3 % (HC), respectively.

New:

The 5-year overall follow-up was 20.8%, but 14.8% in the nESPC and 26.4 % in the HC, respectively.

Old:

The overall survival of patients was comparable between the cohorts. In contrast, death censored graft survival after three and five years was better in the nESPC. 

New: 

The overall survival of patients was comparable between the cohorts. In contrast, death censored graft survival after three and five years was significantly better in the nESPC (Fig 1, Tab 4). As the number of complete 5-year follow-ups was patchy, especially in the nESPC as patients have been transplanted more recently, the 1- and 3-year survival rates were calculated (Table 4). 3-year overall patient and graft survival did not differ significantly. Yet, 3-year death-censored graft survival was significantly better in the new cohort (p = 0.037).

Survival statistics

a. The description of the cox regression analysis can´t be correct. I suppose that the authors intended to state that variables with possible association with five-year patient and graft survival were evaluated in univariable cox regression models and that noticeable values were included in multivariable Cox models (not linear). 

We are very thankful for this important remark. The reviewer is fully correct supposing that we performed a univariable cox regressions and included noticable values in a multivariable Cox regression. 

Old:

Cox regression was used to evaluate the influence on 5-year death-censored graft function and survival and 5-year patient survival [5]. 

New: 

Cox regression was used to evaluate the influence on 5-year death-censored graft function and survival and 5-year patient survival [5]. For multivariable model building, variables with a p-value in the likelihood ratio test > 0.1 were excluded and only variables with a p-value less than 0.05 were further analyzed by multivariable regression. Significant values were further analyzed by multivariable regression. 

The authors have not described the criteria for including variables in the multivariable model and in the tables describing the uni- and multivariable hazard ratios, only one variable is presented in the multivariable model. All variables included in the multivariable model should be included in the table with HR and 95% CI.

We thank the reviewer for this comment. In order to provide a better overview only significant results from the multivariable analysis were mentioned in tables 5, 6 and 7. Multivariable analyses were calculated based on significant results from the univariate analyses. 

b. The log-rank test is a test comparing the survival curves and not the actual 5-year survival. The p-values should consequently be included in the KM-plots and not in the comparison of 5-year survival proportions. 

We thank the esteemed referee for this comment. P-values have been included in the Kaplan-Meier-plots. 

Old: 

New:

Since only a small proportion of patients were followed for five years the 5-year survival estimates are associated with a major uncertainty. 

We thank the esteemed referee for this comment and agree with his point and have added this point in the limitations section. 

Old: 

Additionally, complete 5-year follow-up was relatively small with some gaps in the inclusion period.

New:

Additionally, complete 5-year follow-up was relatively small with some gaps in the inclusion period resulting in limited statistical power.

Comorbidities

a. It is not described which comorbidities that were included and which were not. Table 1 describes the proportions of hypertension, diabetes, arteriosclerosis and hyperlipidemia. Why were these chosen? Why not ischemic heart disease, heart failure or cerebrovascular disease. Several comorbidity scores are developed to describe comorbidity. One of these should be included. Since the study is a single center study including a relatively low number of patients, it should be possible to calculate a comorbidity index for all patients based on hospital records.

We thank the esteemed referee for the comment. We have included arterial hypertension and diabetes mellitus as these two are the two leading comorbidities in patients with chronic kidney diseases (Li PKT, Garcia-Garcia G, Lui SF, Andreoli S, et al. Arch Argent Pediatr 2020). Hyperlipidemia as well as the overweight, represendeted by the BMI, are part of the metabolic syndrom which itself is a risk factor for renal failure (Li PKT, Garcia-Garcia G, Lui SF, Andreoli S, et al. 2020). Arteriosclerosis was included as it is a found universially in uremic patients due to chronic inflammation (Hernández D, Alonso-Titos J, Armas-Padrón AM, Kidney Blood Press Res. 2020). Further, severe arteriosclerosis influences surgical outcome as well as it may cause technical problems. We have included the following in the text: 

Old:

However, only patients who received an ESP kidney were included in the final analysis. 

New: 

However, only patients who received an ESP kidney were included in the final analysis. Recipient data included demographic data, underlying renal disease, most common comorbidities and time of dialysis vintage [2-4]. To further characterize RTx-recipients, comorbidities, including arterial hypertension, diabetes mellitus, hyperlipidemia and arteriosclerosis as well as BMI as a surrogate for overweight were included in the analysis. Arterial hypertension and diabetes mellitus are well described major risk factors in patients with chronic kidney diseases as are hyperlipidemia and overweight [2]. Arteriosclerosis may influence surgical outcome and was therefor included as well [3]. 

Statistical strength

a. A total of 151 patients were included, 62 in the first cohort and 89 in the second cohort. In several analysis this is obviously too few to draw any firm conclusions. For instance, the proportion of intraoperative complications is five-fold higher in the. HC compared to the nESP cohort (3.1 % vs. 15.7 %). The difference is however not significant (p=0.283)

We thank the esteemed referee for this comment and agree with him. We have thankfully included the remark in the limitations section.

Old: 

Further, inclusion periods were quite long with changes occurring concerning procurement procedures such as the change of preservation solution and surgical procedures such as splinting of the ureterocystostomy, allocation criteria and improvement of immunosuppressive therapies. 

New: 

Further, inclusion periods were quite long with changes occurring concerning procurement procedures such as the change of preservation solution and surgical procedures such as splinting of the ureterocystostomy, the intra- and postoperative complications, allocation criteria and improvement of immunosuppressive therapies. These surgical as well as medical changes may influence the outcome of transplantation. Yet, more intra- and postoperative complications were registered in the nESPC still resulting in comparable longterm graft function.

 

Reviewer #2: 

Overall the manuscript is of interest, well written and adequately referenced. The only significant and clinically relevant difference was observed in long-term kidney graft function in favor of the newESP allocation principle. The authors adequately discuss potential differences including a potential role for clinical immunosuppression and prospective matching for HLA class-II antigens. The ESPD program actually started in 2010 randomly assigning kidney from elderly donor as paires with or without matching for HLA-DR.

1) The authors should clearly state whether their 62 newESP recipients were actually part of this ESDP initiative?

We thank the reviewer for this very helpful question. Münster transplant center did indeed participated in the ESDP initiative. However, only one patient was randomized during the recruitment periode. Based on this small numner, we did not take this into account when analysing our dataset and did mention in in the initial manuscript. Since we completely agree wit the editor that this could a potential bias in our results and should be clarified for the readership, we have now included the following statement in our revised manuscript.

Old:

Even though DR-mismatches did not increase in the nESPC at MTC, Dreyer et al.’s suggestions need to be considered as better DR-matching potentially further improves outcome after KTx [1]. 

New:

Even though DR-mismatches did not increase in the nESPC at MTC, Dreyer et al.’s suggestions need to be considered as better DR-matching potentially further improves outcome after KTx [1]. It should be noted that only one patient in the nESPC participated in the Eurotransplant Senior DR compatible Program, thus a potential bias posed by different HLA-DR matching can be excluded.

2) If yes, the better long-term function could be explained by less acute rejection episodes or less steroid resistant episodes and/or less de novo HLa-DR/DQ antibody formation

Münster transplantation center did participate in the ESPD initiative. However, during the recruitement period, only one patient was included. We therefore did not specifically mention this patient as we think that the inclusion of only one patient did not influence the outcome and did not pose a potentital bias. 

 

Reviewer #3: 

The authors presented the results of kidney transplantation in elderly recipients using a new allocation strategy. Although the results are excellent, the authors should better clarify how these organs were allocated: in other words, does this cohort include consecutive patients or patients were selected.

We thank the refereee for this comment and have clarified the paragraph accordingly. Eurotransplant introduced the European Senior Programm (ESP) in 1999. In the beginning however, elderly patients >65 years were listed on the “normal” waiting list and the ESP-waiting list. However, in 2010 both programs became mutually exclusive. After presenting at their nephrologist, potential kidney transplantation recipients present at a Transplant center and discuss their options with a transplantation nephrologist. The transplantation nephrologist introduces both programs and after extensive information, patients have to decide for one waiting list. The ESP tries to reduce cold ischemia times and thereby the associated damages which are made by the ischemia-reperfusion injury and therefore tries to allocate grafts more locally. 

Old:

Since 2010, though, patients have had to pick a program as both programs became mutually exclusive. 

New:

Since 2010, though, patients have had to pick a program as both programs became mutually exclusive. Decision for one program was made after discussion with the consulting transplant nephrologist. 

Could this new allocation strategy penalize the patients who are not included in this group? 

We thank the estimated referee for this very interessting question. As patients liberately decide which waiting list they want to be listed on there is no penelization. Further, one of the aims of the study conducted was to see if patients were penalized or disadvantages by being only listed on the ESP-waiting list with regards to graft function and outcome as well as survival. We could show a trend towards a slightly better, at least not worse outcome in the ESP-cohort and could therefore estimate that patients are not penalized by the choice of the ESP-waiting list. 

Moreover, the authors should better detail in the discussion that the two group were not homogeneous, and that there is a great disparity in terms of cold ischemia time and dialysis vintage, which are well known factors strongly influencing the outcome of kidney transplantation.

We thank the esteemed referee for this comment and acknowledge the concern. We have therefore adjusted the paragraph accordingly. 

Old: 

We hypothesized that the change from ETKAS and ESP to mono-ESP listing would affect waiting time and outcome of KTx in elderly recipients. 

New:

We hypothesized that the change from simultaneously ETKAS and ESP (HC) to mono-ESP (nESP cohort) listing would affect waiting time and outcome of KTx in elderly recipients. Patients in the nESP cohort were older and heavier, suffered more often from hypertension and diabetes, but had experienced shorter dialysis vintage and showed less arteriosclerosis. Yet, more surgical and non-surgical postoperative complications were reported in the nESP cohort.

 

Reviewer #4: Could you detail how patients now decide which waiting list to go on?

We thank the esteemed referee for this comment. After presenting in the transplantation inpatients clinic, patients discuss the choice of waiting list with a specialised transplantations nephrologist and can hereafter deciede which waiting list they want to be listed on. This is a common procedure in the ESP-region. We have modified the paragraph accordingly. 

Old:

Since 2010, though, patients have had to pick a program as both programs became mutually exclusive. 

New:

Since 2010, though, patients have had to pick a program as both programs became mutually exclusive. Decision for one program was made after discussion with the consulting transplant nephrologist.

I have a concern about the comparison groups however....it is not exactly clear what you did. In lines 74- 76 you say you seek to determine whether single listing puts older patients at a disadvantage now compared to when they were wait listed on both lists but unless I have misunderstood you don't seem to answer that question because you say in your final analysis you only include data from the historic cohort who received an ESP kidney (Line 85 - you say only patients who received an esp kidney were included.) Does this mean that your 89 patients in the historic cohort are all those who got an ESP kidney? How many other historic cohort patients got a Tx from the standard list?

We thank the esteemed referee for this comment. Between January 1999 and May 2007 745 patients received kidney transplantations at Muenster Transplantation Center, including 89 elderly patients who received ESP-kidney from elderly donors. In both cohorts, recipients only received ESP-kidneys from donors > 65 years. However, the recipients in the historic cohort were listed on an ESP-waiting list as well as on a common waiting list with all other waiting list patients. The new ESP cohort on the other hand was only listed on the ESP-waiting only including recipients and donors >65 years. The ESP aims for reduced cold ischemia times as well as the associated complications by local allocation without taking HLA-matching into consideration. By this study, we seek to determin the difference between this difference in listing and if the single listing disadvantages the elderly recipients on the exclusive ESP-waiting list. 

Old:

Between January 1999 and May 2007, 89 elderly patients had transplants after being listed on both waiting lists (HC) [4]. 

New:

Between January 1999 and May 2007, 89 elderly patients had transplants after being listed on both waiting lists (HC) [4]. Yet, only recipients of an ESP-kidney were included in the final analysis.

If I have understood the above correctly then you need to provide data on the patients who received a kidney on the standard wait list from the historic cohort. Otherwise the comparison is therefore misleading. How many patients in either cohort on either list died whilst on the list?

We thank the esteemed referee for this comment and acknowledge the concern. Between 2010 – 2014 15 patients awaiting an ESP-kidney died whilst on the ESP-waiting list. For the historic cohort exact numbers are unfortunatelly not reconstructable. However, in a previous puplication from Muenster Transplantation Center, comparing patients on the waiting list with renal transplantation recipients of normal grafts and renal transplantation recipients who received grafts with acute kidney failurewe idendentified a noticably lower survival in patients remaining on the waiting list without receiving neither graft (Schütte-Nütgen K et al., PLoS ONE, 2019). 

Old:

Dialysis vintage was shorter in the nESPC (19.6 ± 21.7 months vs 60.2 ± 28.1 months, p = 0.009). 

New:

Dialysis vintage was significantly shorter in the nESPC (19.6 ± 21.7 months vs 60.2 ± 28.1 months, p = 0.009). In the nESPC 15 patients died while waiting for an ESP-kidney. For the HC exact numbers cannot be reconstructed. In a comparable cohort from Münster Transplant Center, 5-year survival on the waiting list was only 41% [6]. 

A more meaningful comparison would be either the inclusion of all data from the historic cohort ie those who received a transplant on the standard ETKAS as well as the ESP list? Would be of interest to see how they fared compared to those in this cohort who received a Tx on the ESP list and in fact potentially a more useful and meaningful comparison. how long did they wait etc? And/ or a comparison in the new cohort of those who select to go onto the standard wait list v those who elect for the ESP wait list including the demographics. Is there inherent selection bias in those who select one or other list(s)?

We thank the reviewer for this important comment. We agree that an interesting comparision would be to be made between patients who were on the ETKA- and patients who were on the ESP-waiting list. It has already been shown by different groups that ETKAS- as well as ESP-listing result in comparable longterm kindey function (Bahde et al. Ann Transplant 2014, Fabrizii Tranpslantation 2005, Smits Am J Transplantat 2002). However, these results are from the beginnings of the ESP-program. In these years kindey transplantation recipients were listed on boths lists simultanousely. Since 2010, the listing on one list must to be chosen. Further, the allocation policy of the ESP differs in one mayor point: the local allocation in order to shorten cold ischemia times and not primarily take HLA-matching into account. This represents a mayor difference to the allocation policy prior to 2010 and may hence influence the outcome of the renal grafts. 

In case of renal failure, the ETKAS- and the ESP-program are introduced by a transplantation nephrologist and after a sufficient discussion, patients choose one progam, i.e. one waiting list. To our knowledge, this study is the first one to compare a cohort of ESP-recipients which was listed on boths lists, but received an ESP-graft to a cohort which was only listed on the ESP-list receiving an ESP-graft. This is the novelty of this study. Due to the informed consent, we think there is no selection bias in patients for one or the other waiting list and patients are not penalized by being listed on the ESP-list. 

Old: 

This was not achieved at the expense of HLA mismatch, resulting in comparable numbers of primary and overall graft function and rejections. 

New:

This was not achieved at the expense of HLA mismatch, resulting in comparable numbers of primary and overall graft function and rejections. ESP-recipients after single listing therefore do not seem to be penalized by the single listing.

Old:

The listing of younger and elderly patients on a common waiting list probably leads to a longer waiting time for all patients with some patients dying awaiting KTx and with a competition between the age groups [7, 8].

New:

The listing of younger and elderly patients on a common waiting list probably leads to a longer waiting time for all patients with some patients dying awaiting KTx and with a competition between the age groups [7, 8]. In an analysis from our center comparing patients on the waiting list to patients receiving a graft with acute kidney injury of the donor prior to transplantation, we identified a 5-year survival waiting list candidates of 41 % only [6]. Between 2010 and 2014, the inclusion period of the nESPC in this paper, 21.7 % of the patients who died on the waiting list were part of the ESP.

In terms of results: the new cohort compared to the historic cohort are older and heavier, have been on dialysis for a shorter time period, have more hypertension and diabetes but less arteriosclerosis (presumably a factor of shorter dialysis vintage), There are more post op surgical and non surgical complications. Non significant trends towards less DGF, better graft survival and higher levels of rejection.

We thank the reviewer for this excellent remark and have included a paragraph in the respective fields. 

Old: 

We hypothesized that the change from ETKAS and ESP to mono-ESP listing would affect waiting time and outcome of KTx in elderly recipients. 

New:

We hypothesized that the change from simultanously ETKAS and ESP (historic cohort) to mono-ESP (nESP cohort) listing would affect waiting time and outcome of KTx in elderly recipients. Patients in the nESP cohort were older and heavier, suffered more often from hypertension and diabetes, but had experienced shorter times of dialysis vintage and showed less arteriosclerosis. Yet, more surgical and non-surgical postoperative complications were reported in the nESP cohort.

The number for 3 years death censored graft survival is odd in the historic cohort - 33.9%. can you explain?

We are very thankfull for this remark and have corrected this very unfortunate typo. 

Old:

Death-censored graft survival 

(%) 

1 y 93.5 82.3 0.141a

 3 y 93.5 33.9 0.037a

 5 y 93.5 79.3 0.024a

New:

Death-censored graft survival 

(%) 

1 y 93.5 86.2 0.141a

 3 y 93.5 81.6 0.037a

 5 y 93.5 79.3 0.024a

The cox regression analysis - you do univariate and not multivariate analysis presumably due to the small numbers.

We thank the reviewer for this very helpful question. We have performed both univariate and multivariable Cox regression analysis, even though we are aware of the reduced statistical power due to the small number of patients included in the analysis. 

You don't offer an explanation then of why graft survival is better long term? You allude to this in the discussion - ie different surgical techniques, improved immunosuppression regimens, altered preservation/retrieval which takes me back to my first point a better comparison not be of all patients in historic cohort and all patients over 65 on either list in the new cohort?

We thank the reviewer for this valuable comment. We think there is not one explanation for the improved outcome in the nESP-cohort, but rather a sum of changes that have occured within the course of time. However, one mayor difference between the two cohorts is the different listing practice which has been applied to the new cohort. Due to the different listing and the changed allocation policy waiting times and cold ischemia times were reduced. Both factors highly influence the outcome after transplantation. We acknowledge the concerne of the reviewer, but the aim of this study was not the comparisson of same aged patients on the different waiting lists during the same period of time, but to find out if the change in allocation policy penalizes the nESPC. We have adjusted the introduction accordingly in order to emphasize this point. 

Old: 

Therefore, this study aims to determine whether a single listing and the exclusive allocation of elderly ESP-grafts puts elderly patients at a disadvantage compared to a cohort listed on both waiting lists.

New:

Therefore, this study aims to determine whether a single listing and the exclusive allocation of elderly ESP-grafts puts elderly patients at a disadvantage or even penalizes them compared to a cohort listed on both waiting lists.

Figures and tables.

For the Kaplan Meier graphs, I would include the p values and the raw numbers on the curves so it is immediately obvious. Creatinine not keratinin. Similarly need to provide p values and what Asterix represents on the eGFR and creatinine bar charts either in the legend or on figure itself.

We thank the esteemed reviewer for this comment and have modified figures and legends accordingly. 

Old:

New:

Old:

Figure 2 Kidney function using a) creatinine and b) GFR as surrogates during the follow-up period of 60 months

(black HC, grey nESPC) 

HC = historic cohort, nESPC = new European Senior Program cohort

Data are presented as mean ± standard deviation. Student’s t-test was used to test for statistical significance. p-values <0.05 was considered statistically significant.

New:

Figure 2 Kidney function using a) creatinine and b) GFR as surrogates during the follow-up period of 60 months

(black HC, grey nESPC) 

HC = historic cohort, nESPC = new European Senior Program cohort, *** highly significant, p < 0.001

Data are presented as mean ± standard deviation. Student’s t-test was used to test for statistical significance. p-values <0.05 was considered statistically significant. 

Minor points and language/grammatical issues:

We thank the esteemed refree for the comment. p-Values and the raw numbers have been included in the Kaplan-Meier graphs. Kreatinin was changed to creatinine and the legend for *** was added in the legend. 

For the values consistency so quote either 1 or 2 decimal places not a mixture.

We thank the esteemed reviewer for this comment and have modified the numbers accordingly. 

Old:

Weight (26.4 ± 3.9 kg/m2 vs 27.5 ± 3.6 kg/m2, p=0.055), kidney function pre-procurement (serum creatinine 0.95 ± 0.4 mg/dl vs 1.04 ± 0.5 mg/dl, ns); diuresis/last 24 hours (4,074 ± 2,387 ml vs 4,058 ± 2,226 ml, ns) and the number of comorbidities were comparable.

New: 

Weight (26.4 ± 3.9 kg/m2 vs 27.5 ± 3.6 kg/m2, p=0.055), kidney function pre-procurement (serum creatinine 1.0 ± 0.4 mg/dl vs 1.0 ± 0.5 mg/dl, ns); diuresis/last 24 hours (4,074 ± 2,387 ml vs 4,058 ± 2,226 ml, ns) and the number of comorbidities were comparable.

Old:

Sum of comorbidities

(mean ± SD) 2.15 ± 0.9 1.76 ± 0.8 0.009a

New:

Sum of comorbidities

(mean ± SD) 2.2 ± 0.9 1.8 ± 0.8 0.009a

Old:

Creatinine pre-procurement

(mg/dl, mean ± SD) 0.95 ± 0.4 0.95 ± 0.5 0.403a

New:

Creatinine pre-procurement

(mg/dl, mean ± SD) 1.0 ± 0.4 1.0 ± 0.5 0.403a

Old:

Sum of comorbidities 0.88 0.88 0.976a

New:

Sum of comorbidities 0.9 0.9 0.976a

We thank the esteemed referee for the comments and have changed all the below mentioned grammatical issues.

Line 22-24 This sentence in the abstract needs reworded - it does not make sense as it stands

Old: 

This study compared a cohort listed on the general and the ESP waiting lists with a collective exclusively listed on the ESP list concerning different transplantation outcomes.

New:

This study compared a historic cohort of renal transplant recipients being simultanously listed on the general and the ESP waiting lists with a collective exclusively listed on the ESP list 

with regards to surrogates of transplantation outcome.

Line 52 would change offer to supply and after : should be a not A

Old: 

However, recently, the discrepancy between organ demand and offer has further escalated: A decreasing number of organ donations meets an increasing number of patients waiting for KTx which is why an expansion of the donor pool is necessary.

New:

However, recently, the discrepancy between organ demand and supply has further escalated: a decreasing number of organ donations meets an increasing number of patients waiting for KTx which is why an expansion of the donor pool is necessary.

Line 83 An not a (?)

Old: 

The second was a historic cohort (HC) which underwent KTx between January 1999 and May 2007 after parallel listing on the ETKAS and the ESP waiting lists.

New:

The second was an historic cohort (HC) which underwent KTx between January 1999 and May 2007 after parallel listing on the ETKAS and the ESP waiting lists.

Line 101 mycophenolate

Old: 

According to MTC’s protocol, standard immunosuppression consisted of tacrolimus, mycophenolat mofetil and steroids.

New:

According to MTC’s protocol, standard immunosuppression consisted of tacrolimus, mycophenolate mofetil and steroids.

Line 104 co-trimoxazole

Old: 

Cotrimoxazol was administered for Pneumcystis carinii-prophylaxis.

New:

Co-trimoxazole was administered for Pneumcystis carinii-prophylaxis.

Line 121 rejection rather than rejections

Old: 

Death without previous graft failure was regarded as censored. Further, demographic data including underlying renal disease, comorbidities, time of dialysis, HLA mismatch, cold and warm ischemia times, intra- and postoperative complications and biopsy-proven rejections (BPR) were analyzed and compared and their influences on graft outcome tested. 

New:

Further, demographic data including underlying renal disease, comorbidities, time of dialysis, HLA mismatch, cold and warm ischemia times, intra- and postoperative complications and biopsy-proven rejection (BPR) were analyzed and compared and their influences on graft outcome tested. 

Line 124 and 125 as not in

Old: 

Quantitative data was presented in percentages and evaluated using the Student’s t-test. Qualitative data was expressed in means ± standard deviation and ranges and was evaluated using Fisher’s exact test.

New:

Quantitative data was presented as percentages and evaluated using the Student’s t-test. Qualitative data was expressed as means ± standard deviation and ranges and was evaluated using Fisher’s exact test.

Line 126 change noticeable to significant or could use notable but noticeable is incorrect

Old: 

P-values <0.05 were considered noticeable.

New:

P-values <0.05 were considered significant

Line 128 hazard ratios

Old: 

Hazard’s ratio, corresponding 95% confidence intervals and p-values were reported.

New:

Hazard ratios, corresponding 95% confidence intervals and p-values were reported.

Line 129 Noticeable to signifcant or as for line 126

Old: 

Noticeable values were further analyzed by multivariable linear regression

New:

Significant values were further analyzed by multivariable linear regression.

Line 166 primary functions - better to say towards less DGF or tended to be more likely to achieve primary function.

Old: 

However, the nESPC showed a trend towards more primary functions.

New:

However, the nESPC tended to be more likely to achieve primary function.

Line 179 - rejection should not be pleural

Old: 

Furthermore, the main reasons for loss of function were biopsy-proven rejections (21.5% vs 19.1%, ns). 

New:

Furthermore, the main reasons for loss of function were biopsy-proven rejection (21.5% vs 19.1%, ns).

Line 180 - Acute rejection episodes were non significantly increased in the nESPC

Old: 

Acute rejections dominated insignificantly in the nESPC (15.4% vs 13.5%, ns). 

New:

Acute rejection episodes were in tendency increased in the nESPC (15.4% vs 13.5%, ns).

Line 180 add the presence of diabetes

Old: 

Interestingly, only diabetes in recipients was found to significantly influence 5-year graft survival and 5-year death-censored graft survival in the multivariate analysis (p=0.012 and p=0.002, respectively) (Tab 5 and 6).

New:

Interestingly, only the presence of diabetes in recipients was found to significantly influence 5-year graft survival and 5-year death-censored graft survival in the multivariate analysis (p=0.012 and p=0.002, respectively) (Tab 5 and 6). 

Line 191 lymphocele singular

Old: 

Bleeding was the main surgical complication in both cohorts (21.5% vs 12.4%, ns) followed by postoperative lymphoceles in the nESPC (12.3%) and problems of urethral anastomosis in the HC (4.5%).

New:

Bleeding was the main surgical complication in both cohorts (21.5% vs 12.4%, ns) followed by postoperative lymphocele in the nESPC (12.3%) and problems of urethral anastomosis in the HC (4.5%).

Line 222 - The elderly, especially,

Old: 

Especially, the elderly are more susceptible to damages by long CIT and its impact on graft function.

New:

The elderly, especially, are more susceptible to damages by long CIT and its impact on graft function.

Line 225 shorter CIT does not always fully compensate for the effect

Old: 

Nonetheless, short CIT does not always fully antagonize the effect of HLA mismatches.

New:

Nonetheless, shorter CIT does not always fully compensate for the effect of HLA mismatches.

Line 233 and 234 rejection singular

Old: 

HLA mismatch might be one reason for the development of rejections [1, 9]. HLA A and especially HLA DR mismatches correlated with a higher incidence of rejections and graft loss.

New:

HLA mismatch might be one reason for the development of rejection [1, 9]. HLA A and especially HLA DR mismatches correlated with a higher incidence of rejection and graft loss.

---

## [Decision Letter · Decision Letter 1]

1 Jun 2020

PONE-D-20-01629R1

Comparison of Kidney Allograft Survival in the Eurotransplant Senior Program after Changing the Allocation Criteria in 2010 – A Single Center Experience

PLOS ONE

Dear Dr. Reuter,

Thank you for submitting your manuscript to PLOS ONE. After careful consideration, we feel that it has merit but does not fully meet PLOS ONE’s publication criteria as it currently stands. Therefore, we invite you to submit a revised version of the manuscript that addresses the points raised during the review process.

ACADEMIC EDITOR: Please correct the manuscript according to Reviewer 1 suggestions.

We look forward to receiving your revised manuscript.

Kind regards,

Justyna Gołębiewska

Academic Editor

PLOS ONE

Reviewers' comments:

Reviewer's Responses to Questions

**Comments to the Author**

1. If the authors have adequately addressed your comments raised in a previous round of review and you feel that this manuscript is now acceptable for publication, you may indicate that here to bypass the “Comments to the Author” section, enter your conflict of interest statement in the “Confidential to Editor” section, and submit your "Accept" recommendation.

Reviewer #1: All comments have been addressed

Reviewer #2: All comments have been addressed

Reviewer #3: All comments have been addressed

Reviewer #4: All comments have been addressed

2. Is the manuscript technically sound, and do the data support the conclusions?

Reviewer #1: Yes

Reviewer #2: (No Response)

Reviewer #3: (No Response)

Reviewer #4: Yes

3. Has the statistical analysis been performed appropriately and rigorously? 

Reviewer #1: Yes

Reviewer #2: (No Response)

Reviewer #3: (No Response)

Reviewer #4: Yes

4. Have the authors made all data underlying the findings in their manuscript fully available?

Reviewer #1: Yes

Reviewer #2: (No Response)

Reviewer #3: (No Response)

Reviewer #4: (No Response)

5. Is the manuscript presented in an intelligible fashion and written in standard English?

Reviewer #1: Yes

Reviewer #2: (No Response)

Reviewer #3: (No Response)

Reviewer #4: Yes

6. Review Comments to the Author

Reviewer #1: I have reviewed the revised version of the manuscript by Medhorn et al and believe that the authors have improved the manuscript significantly. They have addressed all my remarks, although I still have some minor suggestions for further improvement of the manuscript.

1) Statistics (line 139-141). The authors have tried to describe how their multivariable models were built: “For multivariable model building, variables with a p-value of the likelihood ratio test > 0.1 were excluded and only variables with a p-value less than 0.05 were further analyzed by multivariable regression.” I´m not sure that I understand what the authors mean. What about variables with a p-value between 0.05 and 0.1? If they were included in the multivariable models (as I believe they should), it should have been described as: “For multivariable model building, variables with a p-value of the likelihood ratio test > 0.1 were excluded and only variables with a p-value less than 0.1 were further analyzed by multivariable regression.” Please revise.

2) End points (line 120-121). The limitations of the eGFR should not be mentioned in the methods section but rather in the discussion (where it is actually discussed – line 284-287). This sentence should consequently be removed from the endpoint paragraph of the method section

3) Comorbidities. I accept the authors clarification of why they chose the specific comorbidities for their analyses. However, even if these comorbidities are the most common comorbidities in CKD and development of renal disease. However, they were included in multivariable models as explanatory variable for patient- and graft survival which may be associated with other variables. I think the fact that there are several unknown confounders, as for example other comorbidities, should be mentioned in the limitation section of the discussion.

4) Table 5,6 and 7. I would still prefer to have the p-values of all variables included in the multivariable model, no matter if they are significant or not

Reviewer #2: (No Response)

Reviewer #3: (No Response)

Reviewer #4: All points addressed adequately.

7. PLOS authors have the option to publish the peer review history of their article (what does this mean?). If published, this will include your full peer review and any attached files.

Reviewer #1: No

Reviewer #2: Yes: Johan W de Fijter

Reviewer #3: No

Reviewer #4: No

---

## [Author Response · Author response to Decision Letter 1]

10 Jun 2020

Unfortunately it is impossible to include tables here. Therefore, we kindly refer to the cover letter.

Reviewer #1: I have reviewed the revised version of the manuscript by Mehdorn et al and believe that the authors have improved the manuscript significantly. They have addressed all my remarks, although I still have some minor suggestions for further improvement of the manuscript.

1) Statistics (line 139-141). The authors have tried to describe how their multivariable models were built: “For multivariable model building, variables with a p-value of the likelihood ratio test > 0.1 were excluded and only variables with a p-value less than 0.05 were further analyzed by multivariable regression.” I´m not sure that I understand what the authors mean. What about variables with a p-value between 0.05 and 0.1? If they were included in the multivariable models (as I believe they should), it should have been described as: “For multivariable model building, variables with a p-value of the likelihood ratio test > 0.1 were excluded and only variables with a p-value less than 0.1 were further analyzed by multivariable regression.” Please revise.

We thank the reviewer for this helpful comment and agree that our formulation was difficult to understand. We have modified the paragraph accordingly as the reviewer was fully correct. 

Old:

For multivariable model building, variables with a p-value of the likelihood ratio test > 0.1 were excluded and only variables with a p-value less than 0.05 were further analyzed by multivariable regression. 

New:

For multivariable model building, variables with a p-value of the likelihood ratio test > 0.1 were excluded and only variables with a p-value less than 0.1 were further analyzed by multivariable regression.

2) End points (line 120-121). The limitations of the eGFR should not be mentioned in the methods section but rather in the discussion (where it is actually discussed – line 284-287). This sentence should consequently be removed from the endpoint paragraph of the method section

We thank the reviewer for this valuable comment and have removed the sentence from the methods sections, but emphasized this point in the limitations section. 

Old: 

The primary endpoint was graft function using serum creatinine and estimated glomerular filtration rate (eGFR) calculated according Modification of Diet in Renal Disease Study Group (MDRD) as surrogates [1]. Graft function was chosen as primary endpoint due to the small number of patients included. However, we are aware of the limitations of correlating graft function according to MDRD, as creatinine levels depend on dietary intake and muscle mass, among other factors.

New:

The primary endpoint was graft function using serum creatinine and estimated glomerular filtration rate (eGFR) calculated according Modification of Diet in Renal Disease Study Group (MDRD) as surrogates [1]. Graft function was chosen as primary endpoint due to the small number of patients included.

3) Comorbidities. I accept the authors clarification of why they chose the specific comorbidities for their analyses. However, even if these comorbidities are the most common comorbidities in CKD and development of renal disease. However, they were included in multivariable models as explanatory variable for patient- and graft survival which may be associated with other variables. I think the fact that there are several unknown confounders, as for example other comorbidities, should be mentioned in the limitation section of the discussion.

We thank the reviewer for bringing up this point and fully agree with the point made. We apologize for not having mentioned this before. 

Old:

Unfortunately, we have not found a better way to compare both cohorts. 

New:

Unfortunately, we have not found a better way to compare both cohorts. We have included the most common comorbidities in CKD-patients in order to analyze their influence on graft outcome. However, other comorbidities and other unknown confounders not included in our analysis may also influence outcome after RRT and RTx. 

4) Table 5,6 and 7. I would still prefer to have the p-values of all variables included in the multivariable model, no matter if they are significant or not.

We acknowledge that the reviewer is correct and that the missing data should be included in the final manuskript. We have modified the tables accordingly. 

Old:

Table 5

Cox regression model for 5-year graft survival

Parameters Univariate Multivariable

 HR (95% CI) p-value HR (95% CI) p-value

HC vs nESPC 2.094 (0.831 – 5.278) 0.117 

Recipient age (years) 0.953 (0.866 – 1.050) 0.331 

Recipient sex (male vs female) 1.229 (0.702 – 2.150) 0.471 

Recipient BMI (>25 kg/m2) 2.304 (1.071 – 4.958) 0.033 0.228 (0.036 – 0.288) 0.012 

Time of RRT (months) 1.012 (0.991 – 1.034) 0.274 

Arterial hypertension recipient 0.759 (0.368 – 1.564) 0.455 

Diabetes recipient 1.722 (0.846 – 3.504) 0.134 

Hyperlipidemia recipient 0.877 (0.423 – 1.820) 0.725 

Arteriosclerosis recipient 1.972 (1.059 – 3.701) 0.035 

Cold ischemia time (hours) 0.930 (0.860 – 1.005) 0.066 

Sum of HLA mismatches 0.738 (0.562 – 0.970) 0.029 

Operation time (min) 1.000 (1.000 – 1.000) 0.018 

Warm ischemia time (minutes) 0.985 (0.961 – 1.010) 0.235 

Intraoperative complications (yes vs no) 1.615 (0.638 – 4.091) 0.312 

Postoperative complications (yes vs no) 1.357 (0.764 – 2.411) 0.297 

Length of hospital stay (d) 0.991 (0.974 – 1.009) 0.320 

Donor age (years) 1.011 (0.955 – 1.071) 0.699 

Donor gender (male vs female) 1.402 (0.743 – 2.645) 0.297 

Donor BMI (kg/m2) 1.046 (0.966 – 1.134) 0.270 

Donor creatinine pre-procurement (mg/dl) 0.894 (0.442 – 1.809) 0.756 

Arteriell Hypertension Donor 1.314 (0.671 – 2.576) 0.426 

Diabetes Donor 1.168 (0.528 – 2.582) 0.701 

Arteriosclerosis Donor 0.748 (0.385 – 1.453) 0.391 

Hyperlipidemia Donor 0.614 (0.211 – 1.788) 0.371 

Length of Donor on Intensive Care Unit (d) 1.090 (1.023 – 1.161) 0.007 

PF (yes vs no) 0.075 (0.018 – 0.308) < 0.001 

Biopsy-proven Rejection (yes vs no) 0.826 (0.512 – 1.335) 0.435 

Intraoperative comlications (yes vs no) 1.235 (0.662 – 2.304) 0.507 

Surgical complications (yes vs no) 0.844 (0.569 – 1.252) 0.401 

Non-surgical complications (yes vs no) 0.716 (0.486 – 1.057) 0.093 

Table 5. Cox regression model for 5-year graft survival

HR = hazard ratios, CI = 95% confidence interval. HC = historic cohort, ESP = European Senior Program, BMI = body mass index, HLA = human leukocyte antigen, PF = primary function

New:

Table 5

Cox regression model for 5-year graft survival

Parameters Univariate Multivariable

 HR 

(95% CI) p-value HR 

(95% CI) p-value

HC vs nESPC 2.094 

(0.831 – 5.278) 0.117 

Recipient age 

(years) 0.953 

(0.866 – 1.050) 0.331 

Recipient sex 

(male vs female) 1.229 

(0.702 – 2.150) 0.471 

Recipient BMI 

(>25 kg/m2) 2.304 

(1.071 – 4.958) 0.033 1.813 

(1.175 – 2.799) 0.012

Time of RRT 

(months) 1.012 

(0.991 – 1.034) 0.274 

Arterial hypertension recipient 0.759 

(0.368 – 1.564) 0.455 

Diabetes recipient 1.722 

(0.846 – 3.504) 0.134 

Hyperlipidemia recipient 0.877 

(0.423 – 1.820) 0.725 

Arteriosclerosis recipient 1.972 

(1.059 – 3.701) 0.035 1.371 

(0.886 – 2.121) 0.929

Cold ischemia time 

(hours) 0.930 

(0.860 – 1.005) 0.066 

Sum of HLA mismatches 0.738 

(0.562 – 0.970) 0.029 0.968 

(0.811 – 1.155) 0.449

Operation time 

(min) 1.000 

(1.000 – 1.000) 0.018 1.000 

(1.000 – 1.000) 0.455

Warm ischemia time 

(min) 0.985 

(0.961 – 1.010) 0.235 

Intraoperative complications (yes vs no) 1.615 

(0.638 – 4.091) 0.312 

Postoperative complications (yes vs no) 1.357 

(0.764 – 2.411) 0.297 

Length of hospital stay 

(d) 0.991 

(0.974 – 1.009) 0.320 

Donor age 

(years) 1.011 

(0.955 – 1.071) 0.699 

Donor gender 

(male vs female) 1.402 

(0.743 – 2.645) 0.297 

Donor BMI 

(kg/m2) 1.046 

(0.966 – 1.134) 0.270 

Donor creatinine pre-procurement (mg/dl) 0.894 

(0.442 – 1.809) 0.756 

Arteriell Hypertension Donor 1.314 

(0.671 – 2.576) 0.426 

Diabetes Donor 1.168 

(0.528 – 2.582) 0.701 

Arteriosclerosis Donor 0.748 

(0.385 – 1.453) 0.391 

Hyperlipidemia Donor 0.614 

(0.211 – 1.788) 0.371 

Length of Donor on Intensive Care Unit (d) 1.090 

(1.023 – 1.161) 0.007 1.046 

(0.998 – 1.096) 0.221

PF 

(yes vs no) 0.075 

(0.018 – 0.308) < 0.001 1.400 

(0.917 – 2.135) 0.870

Biopsy-proven Rejection 

(yes vs no) 0.826 

(0.512 – 1.335) 0.435 

Intraoperative complications 

(yes vs no) 1.235 

(0.662 – 2.304) 0.507 

Surgical complications 

(yes vs no) 0.844 

(0.569 – 1.252) 0.401 

Non-surgical complications 

(yes vs no) 0.716 

(0.486 – 1.057) 0.093 

Table 5. Cox regression model for 5-year graft survival

HR = hazard ratios, CI = 95% confidence interval. HC = historic cohort, ESP = European Senior Program, BMI = body mass index, HLA = human leukocyte antigen, PF = primary function

Old:

Table 6

Cox regression model for 5-year death-censored graft survival

Parameters Univariate Multivariable

 HR (95% CI) p-value HR (95% CI) p-value

HC vs nESPC 2.575 (1.016 – 6.527) 0.046 

Recipient age (years) 0.937 (0.848 – 1.035) 0.197 

Recipient sex (male vs female) 1.739 (0.980 – 3.085) 0.058 

Recipient BMI (kg/m2) 2.199 (0.994 – 4.862) 0.052 

Time on dialysis (months) 1.001 (0.989 – 1.034) 0.342 

Arterial hypertension recipient 0.647 (0.315 – 1.330) 0.236 

Diabetes recipient 2.300 (1.119 – 4.731) 0.024 0.0642 (0.336 – 1.311) 0.002

Hyperlipidemia recipient 0.790 (0.372 – 1.681) 0.514 

Arteriosclerosis recipient 2.389 (1.246 – 4.580) 0.009 

Cold ischemia time (hours) 0.947 (0.878 – 1.021) 0.155 

Sum of HLA mismatches 0.744 (0.567 – 0.975) 0.032 

Operation time (min) 1.000 (1.000 – 1.000) 0.097 

Warm ischemia time (minutes) 0.982 (0.957 – 1.008) 0.176 

Intraoperative complications (yes vs no) 1.943 (0.730 – 5.172) 0.184 

Postoperative complications (yes vs no) 1.227 (0.698 – 2.159) 0.478 

Length of hospital stay (days) 0.999 (0.980 – 1.018) 0.918 

Donor age (years) 1.011 (0.953 – 1.072) 0.726 

Donor gender (male vs female) 1.202 (0.611 – 2.365) 0.595 

Donor BMI (kg/m2) 1.050 (0.967 – 1.139) 0.245 

Donor creatinine pre-procurement (mg/dl) 2.575 (1.016 – 6.527) 0.372 

Arteriell Hypertension Donor 1.178 (0.599 – 2.317) 0.643 

Diabetes Donor 0.882 (0.377 – 2.065) 0.773 

Hyperlipidemia Donor 0.763 (0.260 – 2.233) 0.621 

Arteriosclerosis Donor 0.666 (0.347 – 1.277) 0.221 

Length of Stay on Intensive Care Unit (donor) (d) 1.062 (0.997 – 1.132) 0.062 

PF (yes vs no) 0.098 (0.022 – 0.437) 0.002 

Biopsy-proven rejection (yes vs no) 0.695 (0.424 – 1.139) 0.149 

Intraoperative comlications (yes vs no) 0.536 (0.240 – 1.195) 0.127 

Surgical complications (yes vs no) 0.800 (0.436 – 1.469) 0.472 

Non-surgical complications (yes vs no) 0.728 (0.406 – 1.304) 0.286 

Table 6. Cox regression model for 5-year death-censored graft survival

HR = hazard ratios, CI = 95% confidence interval. HC = historic cohort, ESP = European Senior Program, BMI = body mass index, HLA = human leukocyte antigen, PF = primary function

New:

Table 6

Cox regression model for 5-year death-censored graft survival

Parameters Univariate Multivariable

 HR 

(95% CI) p-value HR 

(95% CI) p-value

HC vs nESPC 2.575 

(1.016 – 6.527) 0.046 1.210 

(0.822 – 1.780) 0.276

Recipient age 

(years) 0.937 

(0.848 – 1.035) 0.197 

Recipient sex 

(male vs female) 1.739 

(0.980 – 3.085) 0.058 

Recipient BMI 

(kg/m2) 2.199 

(0.994 – 4.862) 0.052 

Time on dialysis 

(months) 1.001 

(0.989 – 1.034) 0.342 

Arterial hypertension recipient 0.647 

(0.315 – 1.330) 0.236 

Diabetes recipient 2.300 

(1.119 – 4.731) 0.024 1.514 

(0.983 – 2.331) 0.403

Hyperlipidemia recipient 0.790 

(0.372 – 1.681) 0.514 

Arteriosclerosis recipient 2.389 

(1.246 – 4.580) 0.009 1.318 

(0.889 – 1.956) 0.257

Cold ischemia time 

(hours) 0.947 

(0.878 – 1.021) 0.155 

Sum of HLA mismatches 0.744 

(0.567 – 0.975) 0.032 0.925 

(0.795 – 1.078) 0.059

Operation time 

(min) 1.000 

(1.000 – 1.000) 0.097 

Warm ischemia time 

(min) 0.982 

(0.957 – 1.008) 0.176 

Intraoperative complications (yes vs no) 1.943 

(0.730 – 5.172) 0.184 

Postoperative complications (yes vs no) 1.227 

(0.698 – 2.159) 0.478 

Length of hospital stay 

(days) 0.999 

(0.980 – 1.018) 0.918 

Donor age 

(years) 1.011 

(0.953 – 1.072) 0.726 

Donor gender 

(male vs female) 1.202 

(0.611 – 2.365) 0.595 

Donor BMI 

(kg/m2) 1.050 

(0.967 – 1.139) 0.245 

Donor creatinine pre-procurement (mg/dl) 2.575 

(1.016 – 6.527) 0.372 

Arteriell Hypertension Donor 1.178 

(0.599 – 2.317) 0.643 

Diabetes Donor 0.882 

(0.377 – 2.065) 0.773 

Hyperlipidemia Donor 0.763 

(0.260 – 2.233) 0.621 

Arteriosclerosis Donor 0.666 

(0.347 – 1.277) 0.221 

Length of Stay on Intensive Care Unit (donor) (d) 1.062 

(0.997 – 1.132) 0.062 

PF 

(yes vs no) 0.098 

(0.022 – 0.437) 0.002 1.596 

(1.081 – 2.356) 0.622

Biopsy-proven rejection 

(yes vs no) 0.695 

(0.424 – 1.139) 0.149 

Intraoperative complications (yes vs no) 0.536 

(0.240 – 1.195) 0.127 

Surgical complications 

(yes vs no) 0.800 

(0.436 – 1.469) 0.472 

Non-surgical complications 

(yes vs no) 0.728 

(0.406 – 1.304) 0.286 

Table 6. Cox regression model for 5-year death-censored graft survival

HR = hazard ratios, CI = 95% confidence interval. HC = historic cohort, ESP = European Senior Program, BMI = body mass index, HLA = human leukocyte antigen, PF = primary function

Old:

Table 7

Cox regression model for 5-year patient survival

Parameters Univariate Multivariable

 HR (95% CI) p-value HR (95% CI) p-value

HC vs nESPC 1.829 (0.734 – 4.557) 0.195 

Recipient age (years) 0.955 (0.870 – 1.049) 0.337 

Recipient sex (male vs female) 2.100 (1.144 – 3.855) 0.017 

Recipient BMI (<25 kg/m2) 2.225 (0.984 – 5.032) 0.055 

Time on dialysis (months) 1.012 (0.990 – 1.035) 0.287 

Arterial hypertension recipient 0.617 (0.289 – 1.319) 0.213 

Diabetes recipient 2.671 (1.311 – 5.442) 0.007 

Hyperlipidemia recipient 0.989 (0.476 – 2.503) 0.976 

Arteriosclerosis recipient 3.152 (1.620 – 6.132) 0.001 

Cold ischemia time (hours) 0.911 (0.840 – 0.988) 0.025 

Sum of HLA mismatches 0.830 (0.636 – 1.085) 0.173 

Operation time (min) 1.000 (1.000 – 1.000) 0.010 

Warm ischemia time (minutes) 0.987 (0.961 – 1.013) 0.310 

Intraoperative complications (yes vs no) 2.212 (0.823 – 5.945) 0.116 

Postoperative complications (yes vs no) 1.134 (0.652 – 1.972) 0.656 

Length of hospital stay (days) 1.005 (0.986 – 1.025) 0.606 

Donor age (years) 0.976 (0.920 – 1.036) 0.432 

Donor gender (male vs female) 1.113 (0.547 – 2.266) 0.767 

Donor BMI (kg/m2) 0.988 (0.912 – 1.069) 0.761 

Donor creatinine pre-procurement (mg/dl) 1.118 (0.533 – 2.345) 0.769 

Arteriell Hypertension Donor 1.310 (0.689 – 2.489) 0.410 

Diabetes Donor 1.404 (0.635 – 3.106) 0.402 

Hyperlipidemia Donor 2.157 (0.795 – 5.851) 0.131 

Arteriosclerosis Donor 0.793 (0.405 – 1.553) 0.499 

Length of Donor on Intensive Care Unit (d) 1.083 (1.017 – 1.153) 0.013 

PF (yes vs no) 0.272 (0.065– 1.143) 0.076 

Biopsy-proven rejection (yes vs no) 0.707 (0.419 – 1.193) 0.194 

Intraoperative complications (yes vs no) 0.572 (0.297 – 1.102) 0.095 

Postoperative surgical complications (yes vs no) 0.954 (0.634 – 1.436) 0.822 

Non-surgical complications (yes vs no) 0.510 (0.338 – 0.769) <0.001 0.198 (0.015 – 0.272) 0.029

Table 7. Cox regression model for 5-year patient survival 

HR = hazard ratios, CI = 95% confidence interval. HC = historic cohort, ESP = European Senior Program, BMI = body mass index, HLA = human leukocyte antigen, PF = primary function

New:

Table 7

Cox regression model for 5-year patient survival

Parameters Univariate Multivariable

 HR 

(95% CI) p-value HR

(95% CI) p-value

HC vs nESPC 1.829 

(0.734 – 4.557) 0.195 

Recipient age 

(years) 0.955 

(0.870 – 1.049) 0.337 

Recipient sex 

(male vs female) 2.100 

(1.144 – 3.855) 0.017 0.971 

(0.634 – 1.489) 0.422

Recipient BMI 

(<25 kg/m2) 2.225 

(0.984 – 5.032) 0.055 

Time on dialysis 

(months) 1.012 

(0.990 – 1.035) 0.287 

Arterial hypertension recipient 0.617 

(0.289 – 1.319) 0.213 

Diabetes recipient 2.671 

(1.311 – 5.442) 0.007 1.819 

(1.132 – 2.921) 0.090

Hyperlipidemia recipient 0.989 

(0.476 – 2.503) 0.976 

Arteriosclerosis recipient 3.152 

(1.620 – 6.132) 0.001 1.379 

(0.899 – 2.114) 0.958

Cold ischemia time 

(hours) 0.911 

(0.840 – 0.988) 0.025 0.954 

(0.910 – 1.000) 0.272

Sum of HLA mismatches 0.830 

(0.636 – 1.085) 0.173 

Operation time 

(min) 1.000 

(1.000 – 1.000) 0.010 1.000 

(1.000 – 1.000) 0.268

Warm ischemia time 

(min) 0.987 

(0.961 – 1.013) 0.310 

Intraoperative complications 

(yes vs no) 2.212 

(0.823 – 5.945) 0.116 

Postoperative complications 

(yes vs no) 1.134 

(0.652 – 1.972) 0.656 

Length of hospital stay 

(days) 1.005 

(0.986 – 1.025) 0.606 

Donor age 

(years) 0.976 

(0.920 – 1.036) 0.432 

Donor gender 

(male vs female) 1.113 

(0.547 – 2.266) 0.767 

Donor BMI 

(kg/m2) 0.988 

(0.912 – 1.069) 0.761 

Donor creatinine pre-procurement (mg/dl) 1.118 

(0.533 – 2.345) 0.769 

Arteriell Hypertension Donor 1.310 

(0.689 – 2.489) 0.410 

Diabetes Donor 1.404 

(0.635 – 3.106) 0.402 

Hyperlipidemia Donor 2.157 

(0.795 – 5.851) 0.131 

Arteriosclerosis Donor 0.793 

(0.405 – 1.553) 0.499 

Length of Donor on Intensive Care Unit (d) 1.083 

(1.017 – 1.153) 0.013 1.053 

(1.000 – 1.109) 0.351

PF 

(yes vs no) 0.272 

(0.065 – 1.143) 0.076 

Biopsy-proven rejection 

(yes vs no) 0.707 

(0.419 – 1.193) 0.194 

Intraoperative complications

(yes vs no) 0.572 

(0.297 – 1.102) 0.095 

Postoperative surgical complications (yes vs no) 0.954 

(0.634 – 1.436) 0.822 

Non-surgical complications 

(yes vs no) 0.510 

(0.338 – 0.769) <0.001 0.198 

(0.015 – 0.272) 0.029

Table 7. Cox regression model for 5-year patient survival 

HR = hazard ratios, CI = 95% confidence interval. HC = historic cohort, ESP = European Senior Program, BMI = body mass index, HLA = human leukocyte antigen, PF = primary function

---

## [Decision Letter · Decision Letter 2]

22 Jun 2020

Comparison of Kidney Allograft Survival in the Eurotransplant Senior Program after Changing the Allocation Criteria in 2010 – A Single Center Experience

PONE-D-20-01629R2

Dear Dr. Reuter,

We’re pleased to inform you that your manuscript has been judged scientifically suitable for publication and will be formally accepted for publication once it meets all outstanding technical requirements.

Kind regards,

Justyna Gołębiewska

Academic Editor

PLOS ONE

Additional Editor Comments (optional):

Reviewers' comments:

Reviewer's Responses to Questions

**Comments to the Author**

1. If the authors have adequately addressed your comments raised in a previous round of review and you feel that this manuscript is now acceptable for publication, you may indicate that here to bypass the “Comments to the Author” section, enter your conflict of interest statement in the “Confidential to Editor” section, and submit your "Accept" recommendation.

Reviewer #1: All comments have been addressed

2. Is the manuscript technically sound, and do the data support the conclusions?

Reviewer #1: Yes

3. Has the statistical analysis been performed appropriately and rigorously? 

Reviewer #1: Yes

4. Have the authors made all data underlying the findings in their manuscript fully available?

Reviewer #1: Yes

5. Is the manuscript presented in an intelligible fashion and written in standard English?

Reviewer #1: Yes

6. Review Comments to the Author

Reviewer #1: (No Response)

7. PLOS authors have the option to publish the peer review history of their article (what does this mean?). If published, this will include your full peer review and any attached files.

Reviewer #1: Yes: Kristian Heldal

---

## [Editor Report · Acceptance letter]

8 Jul 2020

PONE-D-20-01629R2 

Comparison of Kidney Allograft Survival in the Eurotransplant Senior Program after Changing the Allocation Criteria in 2010 – A Single Center Experience 

Dear Dr. Reuter:

I'm pleased to inform you that your manuscript has been deemed suitable for publication in PLOS ONE. Congratulations! Your manuscript is now with our production department. 

Kind regards, 

on behalf of

Dr. Justyna Gołębiewska 

Academic Editor

PLOS ONE